# VEDiT: Latent Prediction Architecture for Procedural Video Representation Learning

**Han Lin**[*1]**, Tushar Nagarajan**[2]**, Nicolas Ballas**[2]**, Mido Assran**[2]**, Mojtaba Komeili**[2]**,
Mohit Bansal**[1] **& Koustuv Sinha**[2]

[1]UNC Chapel Hill, [2]FAIR, Meta
{hanlincs, mbansal}@cs.unc.edu, koustuvs@meta.com

## Abstract

Procedural video representation learning is an active research area where the objective is to learn an agent which can anticipate and forecast the future given the present video input, typically in conjunction with textual annotations. Prior works often rely on large-scale pretraining of visual encoders and prediction models with language supervision. However, the necessity and effectiveness of extending compute intensive pretraining to learn video clip sequences with noisy text supervision have not yet been fully validated by previous works. In this work, we show that a strong off-the-shelf frozen pretrained visual encoder, along with a well designed prediction model, can achieve state-of-the-art (SoTA) performance in forecasting and procedural planning without the need for pretraining the prediction model, nor requiring additional supervision from language or ASR. Instead of learning representations from pixel space, our method utilizes the latent embedding space of publicly available vision encoders. By conditioning on frozen clip-level embeddings from observed steps to predict the actions of unseen steps, our prediction model is able to learn robust representations for forecasting through iterative denoising —leveraging the recent advances in diffusion transformers (Peebles & Xie, 2023). Empirical studies over a total of five procedural learning tasks across four datasets (NIV, CrossTask, COIN and Ego4D-v2) show that our model advances the strong baselines in long-horizon action anticipation (+2.6% in Verb ED@20, +3.1% in Noun ED@20), and significantly improves the SoTA in step forecasting (+5.0%), task classification (+3.8%), and procedure planning tasks (up to +2.28% in success rate, +3.39% in mAcc, and +0.90% in mIoU). Project page: https://github.com/HL-hanlin/vedit.

## 1 Introduction

Humans regularly perform complex, multi-step *procedural* activities with ease (e.g., cooking a recipe, assembling a piece of furniture). This ability stems from our capacity to recognize, reason about and plan for these activities, which is crucial for developing effective embodied AI systems to perform similar tasks. Towards this, designing systems that can understand procedural activities and predict the next logical steps is an active research problem (Brohan et al., 2023; Chang et al., 2020; Tellex et al., 2011). On the one hand, a large body of prior work on visual representation learning demonstrates the importance of large-scale image or video pretraining for single-step activity understanding (Oquab et al., 2024; Bardes et al., 2024; Zhai et al., 2023; Wang et al., 2023c; Chen et al., 2021; Assran et al., 2023; Xu et al., 2024). On the other hand, encoding sequences of steps (*i.e.*, building a *prediction model*) for future step prediction in videos is a relatively new area of research. Existing procedural video representation learning approaches (Lin et al., 2022; Zhong et al., 2023) typically inherit the same methodology as traditional activity understanding from single short video clip — extending single-clip pretraining to large-scale pretraining on video clip sequences (e.g., in HowTo100M (Miech et al., 2019)) with generic objectives, such as masked step prediction supervised by noisy ASR annotations obtained from narrated videos (Shvetsova et al., 2024) or fixed text knowledge bases like wikiHow (Koupaee & Wang, 2018).

---

[*]Work done during internship at Meta.

However, the necessity and effectiveness of pretraining the prediction model on video clip sequences have not yet been fully validated in these works for two main reasons. First, the dominant single-clip pretraining objectives (e.g., masked token prediction) were designed for feature learning of a short single clip, and are not well aligned to the breadth of downstream procedural tasks (e.g., step forecasting, task classification, procedural planning). Second, pretraining for *sequences* rather than single steps demands a scale of data beyond what is currently available. As a result, current approaches (Lin et al., 2022; Zhong et al., 2023) fall back on text annotations that are often noisy and poorly temporally aligned with the video content (*e.g.*, ASR narrations). Therefore, in this work, we investigate how far an approach can go without requiring extensive pretraining on video clip sequences. Our hypothesis is that learning an efficient prediction model (*i.e.*, transition function) over strong abstract representations from frozen visual encoders offers a compelling alternative to extensive large-scale video clip sequences pretraining for procedural learning tasks.

To this end, we propose our framework **VEDIT - Video Embedding Diffusion Transformer** – a scalable diffusion transformer (DiT, Peebles & Xie (2023))-based prediction model to encode multi-step procedural videos. VEDIT inherits both the diffusion-style training objective and architecture. Specifically, during training, we utilize the latest Flow Matching technique (Esser et al., 2024; Lipman et al., 2023; Goodfellow et al., 2016) for iterative denoising from random Gaussian noise into video clip embeddings. Unlike DiT-based models designed for fine-grained image/video generation (Yang et al., 2024; Esser et al., 2024) which operate at the patch level, our prediction model works as the step/state transition function, utilizing the abstract frame-level representations from frozen visual encoders, operating in latent space (LeCun, 2022). This abstraction allows our model to capture the temporal aspects of the procedural learning task, resulting in the ability to learn an efficient transition function. Crucially, our method *does not require pre-training* as it utilizes existing pre-trained representations, nor does it rely on additional supervision (from text or ASR).

We evaluate our model on five diverse procedural learning tasks across four datasets. (1) On the COIN (Tang et al., 2019) dataset, our model outperforms previous SoTA by a large margin (**+5.0%** for step forecasting and **+3.8%** for task classification), and demonstrates scalable learning as we increase the model size. (2) Our newly proposed VEDIT significantly enhances the overall performance of previous SoTA (Niu et al., 2024) on procedure learning tasks on NIV (Alayrac et al., 2016), CrossTask (Zhukov et al., 2019), and COIN (up to **+2.28%** in success rate, **+3.39%** in mean accuracy, and **+0.90%** in mean IoU), as well as the strong baseline for the Ego4D-v2 (Grauman et al., 2022) long-horizon action anticipation task (**+2.6%** in Verb ED@20, **+3.1%** in Noun ED@20). (3) Finally, we conduct detailed ablation studies on the choice of visual encoders, architecture ablations and large-scale pretraining on video clip sequences, and confirm the effectiveness of each component of our framework design.

In a nutshell, our main contributions can be summarized as follows:

- We propose a procedural video representation learning framework (VEDIT) which leverages diffusion transformers to predict visual representations entirely in the embedding space.
- By combining strong pretrained visual encoders on single video clips with a simple prediction model design, our framework is designed to be trained effectively on a single cross-entropy loss for downstream tasks, eliminating the need for large-scale pretraining on video clip sequences and additional supervision from actions labels or language for learning the prediction model.
- We evaluate VEDIT on five downsteam tasks, including step classification, step forecasting, task classification, procedure planning, and long-term action anticipation across four widely-used benchmark datasets. Our framework outperforms previous SoTAs and baselines by a large margin.

## 2 RELATED WORKS

**Procedural Video Understanding.** Learning procedural knowledge from videos has become an active research area, driven by recent large-scale datasets (Miech et al., 2019; Sener et al., 2022; Afouras et al., 2024; Song et al., 2024) and models trained on them (Niu et al., 2024; Wang et al., 2023a;b; Zhao et al., 2022; Lin et al., 2022; Zhong et al., 2023). These models often rely heavily on large-scale text supervision. For example, DistantSup (Lin et al., 2022) creates text supervision by linking step descriptions from a textual knowledge base (wikiHow) (Koupaee & Wang, 2018) to text narrations from ASR in videos. ProceduralVRL (Zhong et al., 2023) aligns ASR narration embeddings to video representations using strong pretrained image-language models (Radford et al.,

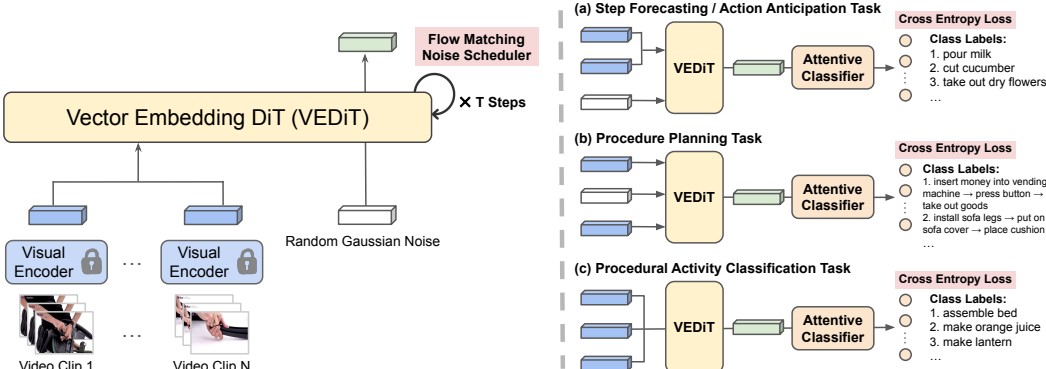

Figure 1: **Overview of our VEDiT training pipeline. Model architecture (left):** We introduce *masked clip-level latent prediction* as our training objective, where we train a Vector Embedding DiT (VEDiT) to iteratively denoise $T$ steps from random gaussian noise with flow matching noise scheduler. **Downstream tasks (right):** We train VEDiT with a light-weight attentive classifier (Bardes et al., 2024) with cross-entropy loss for the following tasks. (a) *Step forecasting / action anticipation task:* predict the embeddings of next unseen clip from observed clips with VEDiT. (b) *Procedure planning task:* predict the embeddings of intermediate unseen clips from observed starting and goal clips with VEDiT. (c) *Procedural activity classification task:* given a sequence of observed video clips, predict the label of the procedural video.

2021). Moreover, prior work uses architectures that necessitate large-scale pretraining on video clip sequences (e.g., self-attention transformers in ProceduralVRL). In contrast, we propose an efficient architecture that learns directly from video, side-stepping the requirement for large-scale language annotations or pretraining on video clip sequences.

**Diffusion Transformers and Flow Matching.** Diffusion models (Song et al., 2020a; Ho et al., 2020; Sohl-Dickstein et al., 2015; Song et al., 2020b) have emerged as a new state-of-the-art architecture for deep generative models. Compared with UNet-based diffusion models (Rombach et al., 2022; Blattmann et al., 2023), recent architectures (Ma et al., 2024c; Chen et al., 2024; Esser et al., 2024; Gao et al., 2024; Yang et al., 2024) designed based on diffusion transformers (Peebles & Xie, 2023) have achieved significant success and scalability in image and video generation. On the other hand, flow matching (Lipman et al., 2023; Ma et al., 2024b; Karras et al., 2022; Nichol & Dhariwal, 2021) has shown great potential as an alternative to DDPM (Nichol & Dhariwal, 2021) for diffusion model noise scheduling. Inspired by these advances, our work introduces a novel modification of the DiT architecture and flow matching for procedural activity learning, that leverages pretrained visual features on single video clips and explicitly models the temporal order of steps.

**Procedural Activity Learning with Diffusion Models.** Prior work has incorporated diffusion into procedural learning in various ways. Some works (Soucek et al., 2024; Black et al., 2024) propose using image- and text-conditioned diffusion-based generation or editing models to generate images of actions and object state changes while preserving the input image scene. These methods primarily use diffusion models as off-the-shelf tools for generating intermediate steps in the pixel space. In contrast, other works (Fang et al., 2023; Shi et al., 2025; Wang et al., 2023b; Zhong et al., 2023) integrate diffusion as a training objective within their model design, predicting the embeddings of unseen target clips based on the embeddings of observed clips. Our work follows this second approach, treating diffusion (flow matching) as a noise scheduler that denoises video embeddings from random noise, and we have designed a procedural learning framework based on the latest DiT architecture. Our framework differs from previous works in three folds: **(1) training supervision:** our model is designed to be trained directly with cross-entropy loss on downstream tasks, without the need for extra language supervision; **(2) prediction model architecture:** instead of using vanilla transformer blocks as the denoising model, we introduce a new Vector Embedding DiT architecture for procedural learning from videos, which is proved to be more effective; **(3) latent embedding generation:** Our model generates unseen video embeddings from a frozen encoder, thereby operating in the latent embedding space.

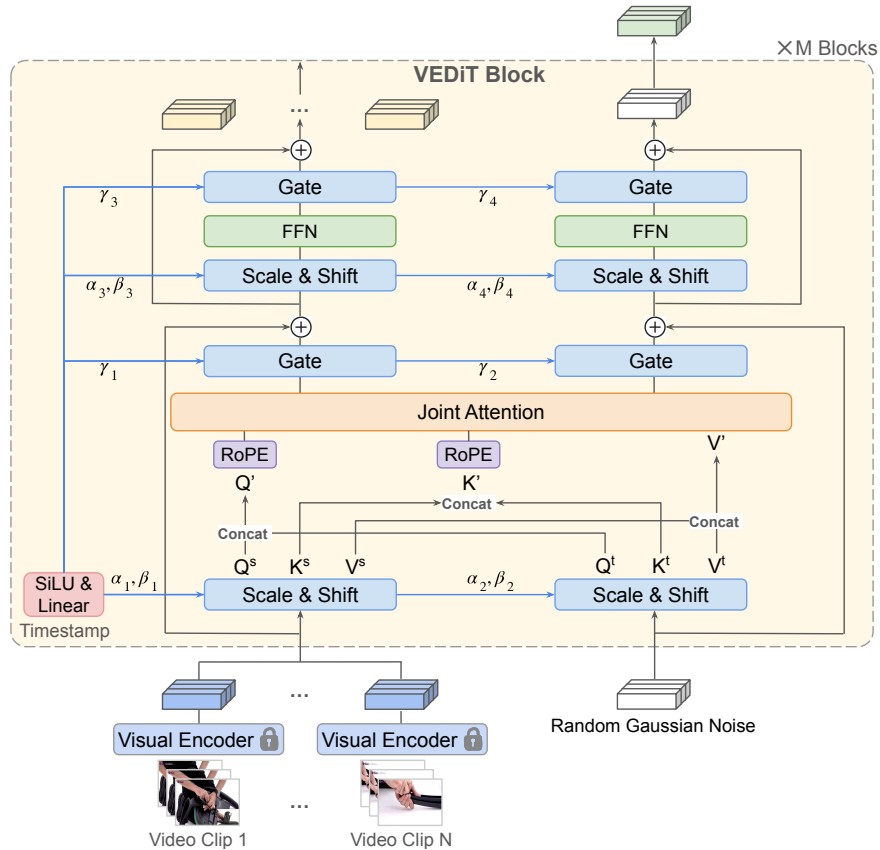

Figure 2: **Vector Embedding Diffusion Transformer (VEDIT) architecture.** During training, our model first uses frozen visual encoders to convert observed video clips into corresponding video embeddings. Then random Gaussian noises are generated as the initial video embeddings of unseen target clips. The DiT-based prediction model processes both seen and target video embeddings in two separate branches, and fuses their information via joint attention blocks where $Q' = \text{Concat}[Q^s, Q^t]$, $K' = \text{Concat}[K^s, K^t]$, $V' = \text{Concat}[V^s, V^t]$. To enable temporal modeling of clips, Rotary positional embeddings (RoPE) is applied to $Q'$ and $K'$ before being input to the attention module. The denoised target clip embeddings are then given as input to the attentive classifier in downstream tasks.

## 3 APPROACH

In this work, we look at three canonical tasks from the procedural representation learning literature, *Step forecasting*, *Procedure Planning*, and *Task classification*, following the setup from Zhong et al. (2023); Niu et al. (2024). Given a series of video clips (or states) from a procedural event (i.e. cooking a dish), the objective is to a) predict the label of the unseen future event (or state) to occur (step forecasting), b) predict the label of unseen events that happened in-between (procedure planning) and c) predict the label of the entire set of events, from a list of probable classes (task classification). Generally, given a sequence of $N$ observable video clip representations $(\boldsymbol{v}_i)$, we aim to learn a procedural state representation $\hat{\boldsymbol{v}}$, which can either capture the information of unseen clips (step forecasting or procedure planning) or a summary information of the task (task classification) using a conditional predictor $\hat{\boldsymbol{v}} = \boldsymbol{\mathcal{F}_\theta}(\{\boldsymbol{v}_i | i \in \mathcal{S}\})$, where $\mathcal{S}$ is a set of observable clips. Using this representation, we then aim to learn a classifier to predict the class labels $(C)$ for the given task $h : \mathbb{R}^{k \times D} \to \mathcal{C}$, where $k$ denotes a set of output embeddings for representation $\hat{\boldsymbol{v}}$ of dimension $D$.

Learning a predictor to predict an unseen clip representation typically requires an extensive training process to learn a rich visual representation *and* temporal information. In this work, we bypass the need of learning visual information by leveraging existing pre-trained encoders $\boldsymbol{v}_i = \boldsymbol{\tau}^*(v_i) \in \mathbb{R}^{k \times D}$, where $v_i$ being the clip in pixel space. Therefore, we focus on learning the temporal transition among clips by operating over the encoder embeddings, $\hat{\boldsymbol{v}} = \boldsymbol{\mathcal{F}_\theta}(\{\boldsymbol{\tau}^*(v_i) | i \in \mathcal{S}\})$, where the predictor needs to generate latent representation $\mathbb{R}^{k \times D}$ embeddings. To generate embeddings with rich visual signals,

we draw inspiration from the diffusion model literature, particularly recent diffusion transformers (DiT) (Peebles & Xie, 2023; Esser et al., 2024). Given their powerful text-conditioned image and video generation capabilities (Yang et al., 2024; Ma et al., 2024a), we adapt their strong conditional generation architecture into a sequential step prediction model for procedural activities. Thus, learning a strong predictor would allow us to generate *unseen* clip embeddings, to enable us to perform the tasks in procedural representation learning.

## 3.1 PRELIMINARY: LATENT DIFFUSION MODEL AND RECTIFIED FLOWS

Given an image $\boldsymbol{x} \in \mathbb{R}^{3 \times H \times W}$ with caption $\boldsymbol{c}$, image Latent diffusion models (LDMs) (Rombach et al., 2022) first use an atoencoder $\mathcal{E}$ to encode the image into latents $\boldsymbol{z}_0 = \mathcal{E}(\boldsymbol{x}) \in \mathbb{R}^{C \times H' \times W'}$, where $C$ represents the number of latent channels, and $H' = H/p$, $W' = W/p$ represent the spatial dimension of the latents, with $p$ denotes as patch size. The forward diffusion process is a fixed diffusion process which adds random noise to the latent variable $\boldsymbol{z}_0$. For example, forward process with Rectified Flows (RFs) (Liu et al., 2023; Lipman et al., 2023; Albergo & Vanden-Eijnden, 2023) is defined as a straight path between the data distribution $\boldsymbol{z}_0$ and a standard normal distribution $\boldsymbol{\epsilon}$ (*i.e.*, $\boldsymbol{z}_t = (1 - t)\boldsymbol{z}_0 + t\boldsymbol{\epsilon}$, where $t \in [0, 1]$). The reverse process in RFs, on the other hand, gradually produces less noise samples starting from $\boldsymbol{z}_1$ to $\boldsymbol{z}_0$ in $T$ denoising steps through a learnable transformer-based denoiser model $\mathcal{F}_{\boldsymbol{\theta}}$ parameterized by $\theta$ and conditioned on caption $\boldsymbol{c}$. In our setup, we transform such denoiser model conditioned on caption/text into a sequential step prediction model for procedural activities conditioned on observed video clips, with explicit temporal order modeling via RoPE (Su et al., 2024).

## 3.2 OUR APPROACH: VECTOR EMBEDDING DIFFUSION TRANSFORMERS (VEDIT)

**Overall Training Pipeline of VEDIT.** The overall training pipeline of our VEDIT is illustrated in Fig. 1 left. Given a set $\mathcal{S}$ that contains $N$ observable (seen) video clips $\{v_1, v_2, ..., v_N\}$, where each video clip $v_i \in \mathbb{R}^{K \times 3 \times H \times W}$ contains $K$ frames, we first apply a frozen visual encoder $\boldsymbol{\tau}^*(.)$ to derive the corresponding video embeddings for each clip $\boldsymbol{v}_i = \boldsymbol{\tau}^*(v_i)$. Next, random Gaussian noises are generated as the initial video embeddings of unseen clips $\{\tilde{\boldsymbol{v}}_j | j \in \mathcal{T}\}$, where $\mathcal{T}$ represents the target set. Then we design VEDIT as the learnable prediction model $\mathcal{F}_{\boldsymbol{\theta}}$ which predicts the unseen target video clip embeddings $(\hat{\boldsymbol{v}})$ conditioned on all seen video embeddings: $\hat{\boldsymbol{v}}_j = \mathcal{F}_{\boldsymbol{\theta}}(\{\boldsymbol{v}\}_i, \tilde{\boldsymbol{v}}_j), i \in \mathcal{S}; j \in \mathcal{T}$. This prediction model is then trained using iterative denoising (Ho et al., 2020) over $T$ steps, with diffusion timestamps sampled from the Flow Matching Euler Discrete Scheduler (Esser et al., 2024). Unlike DiT, which is based-on pixel-level and text-conditioned generation, our model is designed to predict abstract video features in the procedural activity, based on observed video clips. Additionally, unlike previous works (Lin et al., 2022; Zhong et al., 2023) for procedural activity understanding, our method does not use extra language supervision such as ASR or textual knowledge base (*e.g.*, wikiHow) that aligns visual embeddings with text.

**Training Objective of VEDIT.** A key innovation and distinction of our method, compared with previous approaches to video embedding prediction for downstream tasks (Zhong et al., 2023; Lin et al., 2022)—which typically rely on a combination of multiple loss functions (such as video embedding reconstruction loss and video-language matching loss) for supervision—is that our pipeline can be effectively trained with a single cross-entropy loss. Unlike previous works that enforce alignment of the predicted video embeddings with noisily annotated language descriptions, using a single cross-entropy loss allows the optimization target to align more effectively with downstream datasets.

**Choice of Visual Encoder.** Previous works on procedural learning (Lin et al., 2022; Zhong et al., 2023; Niu et al., 2024) typically use clip-level features as abstracted visual representation for each video clip, which can result in a loss of detail. Conversely, DiT models for image and video generation (Yang et al., 2024; Esser et al., 2024; Peebles & Xie, 2023) are designed for patch-level generation with a focus on fine-grained visual details, but this comes at the cost of higher computational demands. To ensure that our model can process videos with multiple clips, encode sufficient visual information, and avoid excessive computational costs, we explore our model with diverse CLIP- and SSL-based encoders that output visual features at the clip-, frame-, and patch-levels. We found empirically that using the `[CLS]` tokens of SigLIP (Zhai et al., 2023) from 16 uniformly

sampled frames in each clip stands out as the strongest visual representation. An ablation study of visual encoders is discussed in Sec. 4.3.

**VEDIT Model Architecture Design.** Fig. 2 visualizes the components of each VEDIT block. Our architecture is derived from DiT (Peebles & Xie, 2023), which has a two-branch architecture with one *query* branch (typically text) tasked to condition the *target* branch (vision) through adaptive layernorm (Perez et al., 2018). In our work, we utlize the *query* branch to process the observable (or seen) video encoder embeddings $\{\boldsymbol{v}_i | i \in \mathcal{S}\}$, which conditions the *target* branch that operates on unseen, noisy embeddings $\{\tilde{\boldsymbol{v}}_j | j \in \mathcal{T}\}$ through adaptive layernorm, which gets iteratively updated through denoising. The query branch is further conditioned by using the timestamp $t \in T$ sampled from the noise scheduler, along with its usual application of determining the scale of noise. Unlike DiT, the information of the query branch and the target branch are fused together using joint attention before being processed independently through feed-forward layers without weight sharing. Lastly, to enable temporal modeling of clips, Rotary positional embeddings (RoPE) (Su et al., 2024) is applied to the input immediately prior to the joint attention module. By utilizing these mechanisms, VEDIT allows us to learn unseen video embeddings, starting from noise, by conditioning on the observed video encoder representations. More ablations on design choices are provided in Appendix A.2.

# 4 EXPERIMENTS

In this section, we first introduce the evaluation datasets and the implementation details of our model in Sec. 4.1. We then compare our method with SOTAs on five downstream tasks across four datasets in Sec. 4.2. Finally, we present ablation studies on the visual encoders, as well as the necessity of pretraining on video clip sequences in Sec. 4.3. More ablations on VEDIT architecture design is provided in Appendix A.2.

## 4.1 EXPERIMENTAL SETUP

**Evaluation Tasks and Datasets.** We evaluate our method on five downstream tasks across four datasets. COIN (Tang et al., 2019) contains 476 hours of YouTube videos covering 180 tasks and 778 unique steps of daily activities. Following (Zhong et al., 2023; Lin et al., 2022), we evaluated our model on two tasks: step forecasting and task classification (see Fig. 1 for details). For Ego4D-v2 (Grauman et al., 2022), we focus on the long-term action anticipation benchmark, which aims to predict the next $Z = N - t$ future action classes [(verb$_1$, noun$_1$), (verb$_2$, noun$_2$), ..., (verb$_Z$, noun$_Z$)] given an input video up to timestamp $t$. This forecasting benchmark contains 243 hours of videos with a total of 3472 annotated clips. In addition, we utilize NIV (Alayrac et al., 2016), CrossTask (Zhukov et al., 2019), and COIN datasets to evaluate the procedure planning task (Chang et al., 2020), which can be seen as a variant of step forecasting task that aims at predicting intermediate action steps given the observed start and goal video clips. Specifically, CrossTask dataset contains 2750 videos covering 18 tasks and 133 actions, and NIV dataset contains 150 videos with 5 tasks and 48 actions. Following (Niu et al., 2024), we report the results with prediction horizon $T \in \{3, 4\}$.

**Evaluation Metrics.** For COIN step forecasting and task classification tasks, we use top-1 classification accuracy of the predicted step/task as the evaluation metric following DistantSup (Lin et al., 2022). For Ego4D-v2 long-horizon anticipation task, we use the default edit distance (ED) metric, which is computed as the Damerau-Levenshtein distance (Damerau, 1964) over sequences of predicted verbs or nouns. Following (Grauman et al., 2022), we report the minimum edit distance at $Z = 20$ (ED@20) for $K = 5$ predicted sequences on the validation set. In addition, for procedure planning tasks on NIV, CrossTask, and COIN, we evaluate the models on three metrics, including success rate (SR), mean Accuracy (mAcc) and mean Intersection over Union (mIoU) following previous works (Chang et al., 2020; Niu et al., 2024; Zhao et al., 2022).

**Implementation Details.** Our default VEDIT architecture contains 12 transformer blocks, with a hidden size of 2048 and attention head dimension of 64. During training, we apply classifier-free guidance with a scale of 7 and denoise the diffusion model for 24 steps using the Flow Matching Euler Discrete Scheduler (Esser et al., 2024). For COIN step forecasting and task classification tasks, we use a scheduled learning rate linearly increases from $5 \times 10^{-6}$ to $5 \times 10^{-5}$ during the first 3 epochs, and then decays to $5 \times 10^{-7}$ following a cosine schedule, with a total of 30 epochs. For long-horizon anticipation and the procedure planning tasks, following the same training setting in previous

works (Grauman et al., 2022; Niu et al., 2024), we train the model for 100 and 500 epochs respectively. Together with VEDIT, we train task specific attentive classifiers $h : \mathbb{R}^{k \times D} \to C$ (Bardes et al., 2024), which is an attentive pooler over $k$ output embeddings followed by a single linear layer.

## 4.2 MAIN RESULTS

**Step Forecasting and Task Classification.** In Table 1, we demonstrate the effectiveness of our VEDIT design on the COIN step forecasting and task classification tasks. Firstly, we use the pre-trained TimeSformer (Bertasius et al., 2021) visual encoder as $\tau^*$, as used in ProceduralVRL (Zhong et al., 2023), the previous state-of-the-art in these tasks. We combined the TimeSformer encoder with VEDIT designed with joint attention, to arrive at the model TimeSformer+VEDIT. This model achieves improvements of 2.2% and 0.6% in top-1 accuracy on the step forecasting and task classification tasks. Next, using SigLIP (Zhai et al., 2023) as $\tau^*$ with VEDIT yields additional gains of **3.1%** and **3.2%** on these two tasks. It is worth noting our methods *does not require any large-scale pretraining on video clip sequences*, which proves the effectiveness of using strong language-aligned single-clip pretrained representations. Additionally, our method *does not require explicit text supervision* (*i.e.*, unsupervised) compared to baselines DistantSup and ProceduralVRL, which are trained with explicit language matching loss (*i.e.*, ASR or ASR+wikiHow). Furthermore, we observe linear *scalability* of VEDIT on Step Forecasting task, leading to improved numbers with increasing number of model parameters (Appendix A.2.4).

**Procedure Planning Task.** We further evaluate VEDIT on procedure planning results on the NIV, COIN, and CrossTask datasets with horizons $T \in \{3, 4\}$ in Table 2. Specifically, we build upon the previous SoTA model, SCHEMA (Niu et al., 2024), by replacing their vanilla transformer blocks in the state decoder and step decoder with our VEDIT blocks. To ensure a fair comparison, we use the same number of transformer blocks (*i.e.*, 2 blocks) with identical hidden dimensions, attention heads, and we strictly adhere to their training and evaluation hyperparameters and setups without any changes. We report the mean and standard deviation of SR, mAcc, and mIoU for our results as well as our replication of SCHEMA averaged over 10 runs.

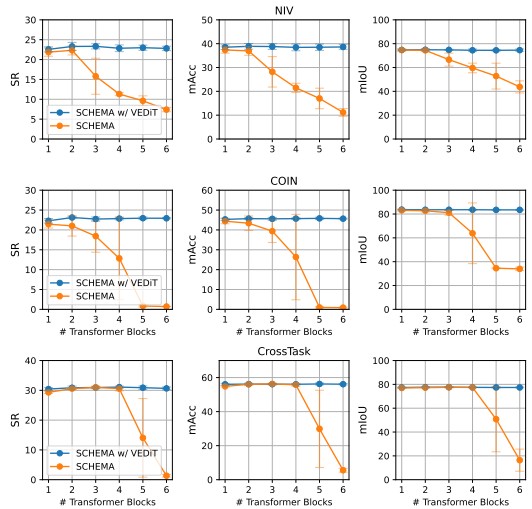

Figure 3: SCHEMA w/ VEDIT is more stable than SCHEMA w/ vanilla transformer as we increase the number of transformer blocks.

As shown in Table 2, SCHEMA with VEDIT consistently outperforms the original SCHEMA method on NIV (gains of 0.97%-2.28% for SR, 1.92%-3.39% for mAcc, and 0.49%-0.90% for mIoU) and COIN (gains of 2.21%-5.89% for SR, 2.31%-7.07% for mAcc, and 0.86%-2.57% for mIoU). Additionally, our VEDIT achieves better average performance on the CrossTask dataset. Moreover, as plotted in Fig. 3, we also observe better stability of VEDIT over vanilla transformer blocks as we scale up the number of transformer blocks.

**Long-Horizon Action Anticipation.** In Table 3, we evaluate our model on the Ego4D-v2 long-horizon action anticipation task. We introduce a new baseline by replacing the SlowFast (Feichtenhofer et al., 2019) visual encoder in the Ego4D baseline model with SigLIP, while keeping the prediction model (*i.e.*, `slowfast_trf_v2`) unchanged. For a fair comparison, we initialize VEDIT with the same number of transformer blocks and hidden dimension size as the Ego4D baseline transformer prediction model. Using SigLIP as visual encoder, VEDIT outperforms the Ego4D baseline in both Verb ED@20 (Ours: 0.697 v.s. Ego4D Baseline: 0.718, lower is better) and noun ED@20 (Ours: 0.711 v.s. Ego4D Baseline: 0.742, lower is better). In addition, while not directly comparable, PaMsEgoAI (Ishibashi et al., 2023), achieves lower ED metrics by introducing several enhancements, including an ensemble of SlowFast and SlowFast-CLIP models, label smoothing to relax order constraints for future actions, and constraining the (verb, noun) classes based on

| Model | Pretraining Supervision | Pretrain Data | Step Forecasting | Task Classification |
|---|---|---|---|---|
| Random Guess | N/A | N/A | 0.1 | - |
| SlowFast (Feichtenhofer et al., 2019) | Supervised: action labels | Kinetics | 25.6 | 71.6 |
| S3D (Xie et al., 2018) | Unsupervised: ASR w. MIL-NCE | HT100M | 28.1 | 70.2 |
| ClipBERT (Lei et al., 2021) | Supervised: captions | COCO+VG | - | 65.4 |
| VideoCLIP (Xu et al., 2021) | Unsupervised: ASR | HT100M | - | 72.5 |
| TSN (RGB + Flow) (Tang et al., 2019) | Supervised: action labels | Kinetics | - | 73.4 |
| TimeSformer (Bertasius et al., 2021) | Supervised: action labels | Kinetics | 34.7 | 83.5 |
| TimeSformer (Bertasius et al., 2021) | Unsupervised: ASR w. MIL-NCE | HT100M | 34.0 | 85.3 |
| DistantSup (Lin et al., 2022) | Unsupervised: ASR + wikiHow | HT100M | 39.4 | 88.9 |
| ProceduralVRL (Zhong et al., 2023) | Unsupervised: ASR | HT100M | 46.8 | 90.8 |
| Ours: TimeSformer + VEDIT | N/A | N/A | 48.7 | 91.1 |
| Ours: SigLIP (Zhai et al., 2023) + VEDIT | N/A | N/A | **51.8** | **94.6** |

Table 1: Step forecasting and task classification results on COIN (Tang et al., 2019) dataset. We compare our method with a set of strong baselines as well as SOTA methods. Top-1 accuracies are reported. We **bold** and underline the best and the second best models in each task respectively.

| Datasets | Models | $T = 3$ | | | $T = 4$ | | |
|---|---|---|---|---|---|---|---|
| | | SR (↑) | mAcc (↑) | mIoU (↑) | SR (↑) | mAcc (↑) | mIoU (↑) |
| NIV | Random | 2.21 | 4.07 | 6.09 | 1.12 | 2.73 | 5.84 |
| | DDN (Chang et al., 2020) | 18.41 | 32.54 | 56.56 | 15.97 | 27.09 | 53.84 |
| | Ext-GAIL (Bi et al., 2021) | 22.11 | 42.20 | 65.93 | 19.91 | 36.31 | 53.84 |
| | P³IV (Zhao et al., 2022) | 24.68 | 49.01 | 74.29 | 20.14 | 38.36 | 67.29 |
| | EGPP (Wang et al., 2023a) | 26.05 | **51.24** | 75.81 | 21.37 | **41.96** | **74.90** |
| | SCHEMA (Niu et al., 2024) | 27.93 | 41.64 | 76.77 | 23.26 | 39.93 | 76.75 |
| | SCHEMA† | 26.66±2.27 | 39.94±2.79 | 75.58±1.47 | 22.32±1.15 | 36.96±1.84 | 74.39±1.13 |
| | SCHEMA w/ VEDIT | **28.94±1.07** | 43.33±0.90 | **76.48±0.62** | **23.29±0.44** | 38.88±1.19 | 74.88±0.89 |
| | (Ours) | (2.28↑) | (3.39↑) | (0.90↑) | (0.97↑) | (1.92↑) | (0.49↑) |
| COIN | Random | <0.01 | <0.01 | 2.47 | < 0.01 | < 0.01 | 2.32 |
| | Retrieval | 4.38 | 17.40 | 32.06 | 2.71 | 14.29 | 36.97 |
| | DDN (Chang et al., 2020) | 13.90 | 20.19 | 64.78 | 11.13 | 17.71 | 68.06 |
| | P³IV (Zhao et al., 2022) | 15.40 | 21.67 | 76.31 | 11.32 | 18.85 | 70.53 |
| | EGPP (Wang et al., 2023a) | 19.57 | 31.42 | **84.95** | 13.59 | 26.72 | **84.72** |
| | SCHEMA (Niu et al., 2024) | 32.09 | 49.84 | 83.83 | 22.02 | 45.33 | 83.47 |
| | SCHEMA† | 26.38±3.66 | 43.08±4.28 | 81.49±1.70 | 21.00±2.56 | 43.37±3.64 | 82.70±1.08 |
| | SCHEMA w/ VEDIT | **32.27±0.44** | **50.15±0.31** | 84.07±0.38 | **23.11±0.27** | **45.68±0.52** | 83.56±0.45 |
| | (Ours) | (5.89↑) | (7.07↑) | (2.57↑) | (2.11↑) | (2.31↑) | (0.86↑) |
| CrossTask | Random | <0.01 | 0.94 | 1.66 | < 0.01 | 0.83 | 1.66 |
| | Retrieval | 8.05 | 23.30 | 32.06 | 3.95 | 22.22 | 36.97 |
| | DDN (Chang et al., 2020) | 12.18 | 31.29 | 47.48 | 5.97 | 27.10 | 48.46 |
| | Ext-GAIL (Bi et al., 2021) | 21.27 | 49.46 | 61.70 | 16.41 | 43.05 | 60.93 |
| | P³IV (Zhao et al., 2022) | 23.34 | 49.96 | 73.89 | 13.40 | 44.16 | 70.01 |
| | PPDP (Wang et al., 2023b) | 26.38 | 55.62 | 59.34 | 18.69 | **52.44** | 62.38 |
| | EGPP (Wang et al., 2023a) | 26.40 | 53.02 | 74.05 | 16.49 | 48.00 | 70.16 |
| | SCHEMA (Niu et al., 2024) | 31.83 | 57.31 | 78.33 | 19.71 | 51.85 | 74.46 |
| | SCHEMA† | 30.57±0.38 | 56.02±0.32 | **77.60±0.25** | 20.26±0.33 | 51.93±0.17 | 74.51±0.25 |
| | SCHEMA w/ VEDIT | **31.08±0.31** | **56.15±0.57** | 77.54±0.35 | **20.42±0.24** | 52.26±0.51 | **74.76±0.29** |
| | (Ours) | (0.51↑) | (0.13↑) | (0.06↓) | (0.16↑) | (0.33↑) | (0.25↑) |

Table 2: Procedure planning results on NIV (Alayrac et al., 2016), COIN (Tang et al., 2019), and CrossTask (Zhukov et al., 2019) datasets with prediction horizon $T \in \{3, 4\}$. SCHEMA†: our replication of their method averaged over 10 runs. The best numbers are **bolded**. Our improvement over SCHEMA baseline is colored in blue.

word co-occurrence. Some studies (Zhao et al., 2024; Pei et al., 2024; Huang et al., 2023) have found that combining vision models with the strong planning capabilities of LLMs can achieve good performance, particularly for long-horizon action anticipation tasks. Therefore, integrating our method with an LLM could be a promising future direction.

## 4.3 ABLATIONS

### 4.3.1 WHICH VISUAL ENCODER WORKS BEST?

To test the impact of visual encoders on procedural activity understanding from instructional videos, we train VEDIT with 3 blocks for 10 epochs with different visual encoders. We include strong CLIP-based and self-supervised (SSL) encoders, including SigLIP (ViT-SO400M/14@384) (Zhai et al., 2023), V-JEPA (Bardes et al., 2024), DINOv2 (Oquab et al., 2024), and VideoMAE Tong et al.

| Method | Encoder | Prediction Model | ED@5 (↓) Verb | ED@5 (↓) Noun | ED@20 (↓) Verb | ED@20 (↓) Noun |
|---|---|---|---|---|---|---|
| Ego4D Baseline (Grauman et al., 2022) | SlowFast | Transformer | - | - | 0.745 | 0.779 |
| Ego4D Baseline (Grauman et al., 2022) | SigLIP | Transformer | 0.703 | 0.736 | 0.718 | 0.742 |
| PaMsEgoAI (Ishibashi et al., 2023) | SlowFast + CLIP | Concat + Transformer | - | - | 0.670 | 0.629 |
| Ours | SigLIP | VEDIT | 0.677 | 0.711 | 0.697 | 0.711 |

Table 3: Comparison of methods on the validation set of Ego4D (Grauman et al., 2022) long-term action anticipation challenge. Edit distance (ED) metrics are reported at prediction horizon 5 and 20.

| Model | Architecture | Pretrain Data | CLIP/SSL | Token | Step Forecasting | Task Classification |
|---|---|---|---|---|---|---|
| DINOv2 | ViT-Giant/14@224 | LVD-142M (Oquab et al., 2024) | SSL | [CLS] | 47.03 | 90.89 |
| VJEPA | ViT-Huge/16@224 | VideoMix2M (Bardes et al., 2024) | SSL | Patch | 47.24 | 86.13 |
| VJEPA | ViT-Huge/16@384 | VideoMix2M (Bardes et al., 2024) | SSL | Patch | 48.23 | 87.14 |
| VideoMAE | ViT-Huge/16@224 | K400 (Kay et al., 2017) | SSL | Patch | 44.78 | 83.40 |
| SigLIP (default) | ViT-SO400M/14@384 | WebLI (Chen et al., 2023) | CLIP | [CLS] | **50.05** | **94.38** |

Table 4: Ablation on frozen video encoders on COIN (Tang et al., 2019) step forecasting and procedural activity classification tasks. Top-1 accuracies are reported. We **bold** and underline the best and the second best models in each task respectively.

(2022). For V-JEPA and VideoMAE, we provide patch tokens to VEDIT, while for DINOv2 and SigLIP, we provide [CLS] tokens. We evaluate the model trained with different encoders on COIN for step forecasting and task classification tasks.

As we observe from Table 4, SigLIP outperforms SSL-based encoders. Among the SSL-based encoders, VJEPA ViT-H 384 and DINOv2 performs comparably than the baselines for step forecasting and task classification task respectively. SigLIP outperforms both on a large margin, especially in Task classification, highlighting the need of language-aligned rich visual representations for stronger procedural activity understanding.

### 4.3.2 IS PRE-TRAINING NECESSARY?

Instead of training directly on downstream datasets (*i.e.*, COIN), previous works (Zhong et al., 2023; Lin et al., 2022) undergo large-scale pretraining on video clip sequences from publicly available video datasets, such as HowTo100M (Miech et al., 2019). In this section, we question whether such clip-sequence pretraining is necessary. Specifically, we compare our model trained directly on COIN with a variant that includes additional clip-sequence pretraining on HowTo100M dataset. Following the reicpe of Zhong et al. (2023), we set the total number of clips to 9, randomly mask out the video embedding of one clip, and use the masked clip embedding reconstruction as the training objective. Additionally, instead of relying on the noisy automatic speech recognition (ASR) annotations, we use the starting and ending timestamps of each video clip processed and filtered by HowToCaption (Shvetsova et al., 2024). During pretraining, we employ the AdamW (Loshchilov & Hutter, 2019) optimizer, with the learning rate linearly increasing from $1 \times 10^{-5}$ to $1 \times 10^{-4}$ during the first 0.5 training epochs, and then remaining constant for a total of 30 epochs. The pretraining is conducted on 128 H100 GPUs with a total batch size of 1024, and takes 2 days and 4.5 days for the 165M and 1.77B VEDIT models respectively (see Table 9 for model architecture details).

Surprisingly, we observe only marginal improvement with significant clip-sequence pretraining (Table 5). Pretraining on video clip sequences only provides an additional 0.3% boost in top-1 accuracy for the step forecasting task. We hypothesize this limited effectiveness of clip-sequence pretraining may stem from two factors: (1) pretraining dataset is relatively noisy, leading to a distribution gap with the downstream COIN dataset, and (2) the pretraining objective in (Zhong et al., 2023) may not be optimal. Exploring better clip-sequence pretraining objectives that can generalize well across different downstream tasks is left for future work.

## 5 LIMITATIONS

One potential limitation of our model with multi-step denoising is that it sacrifices efficiency for performance. Additionally, it is not specifically designed for real-time inference, which is a parallel

| Model | Pretraining Supervision | Pretrain Data | Step Forecasting | Task Classification |
|---|---|---|---|---|
| SigLIP + VEDiT | N/A | N/A | 51.8 | **94.6** |
| SigLIP + VEDiT | Unsupervised: HowToCaption | HT100M | **52.1** | **94.6** |

Table 5: Effect of large-scale clip-sequence pretraining. We compare VEDiT without large-scale pretraining with a variant that's pretrained on 1.16M videos from HowTo100M (Miech et al., 2019) dataset, using temporal information from HowToCaption (Shvetsova et al., 2024). Top-1 accuracies are reported.

| # Denoising Steps | 1 | 4 | 8 | 12 | 16 | 20 | 24 |
|---|---|---|---|---|---|---|---|
| Training Clock Time (sec.) | 0.41 | 0.49 | 0.63 | 0.82 | 0.94 | 1.09 | 1.24 |
| Inference Clock Time (sec.) | 0.40 | 0.47 | 0.56 | 0.66 | 0.75 | 0.84 | 0.93 |
| Training GPU Memory (GB) | 21.8 | 21.9 | 22.0 | 22.0 | 22.1 | 22.2 | 22.3 |
| Inference GPU Memory (GB) | 15.7 | 15.7 | 15.8 | 15.8 | 15.8 | 15.8 | 15.9 |

Table 6: Training and inference clock time and GPU memory of VEDiT as we increase the number of denoising steps.

topic to our paper and typically involves techniques such as model distillation, quantization, and hardware-level optimization. We leave the exploration of this direction for future work.

Here, we provide some analysis by evaluating the training and inference time as well as GPU memory usage of our model with varying numbers of denoising steps. Specifically, we conducted this experiment using the VEDiT architecture with 696M trainable parameters. We measured the clock time and GPU memory required to run inference on the model using 1 COIN video consisting of 8 clips, with gradient checkpointing enabled.

As shown in the table below, increasing the number of denoising steps from 1 to 24 results in only a $1.32\times$ increase in inference time and a $2.02\times$ increase in training time. This efficiency is partially because of the adoption of efficient scalable dot product attention in each VEDiT block. Moreover, because we employ gradient checkpointing to optimize GPU memory usage, GPU memory usage remains nearly constant without significant variation.

## 6 CONCLUSION

In this work, we demonstrate that carefully designed predictive models learned on top of single-clip pretrained visual representations can achieve state-of-the-art performance on procedural learning tasks across the COIN, CrossTask, NIV, and Ego4D datasets, including step forecasting, procedural activity classification, procedure planning, and long-term action anticipation. Notably, we achieve these results without pretraining the prediction module, instead learning it directly from the end tasks. This contrasts with previous works, which often require computationally expensive pretraining of the predictor, sometimes with additional supervision. Our findings suggest that further research is needed to improve clip-sequence pretraining for procedural activities. Specifically, exploring ways to better align pretraining tasks with downstream tasks could help fully leverage the benefits of pretraining. Moreover, the data distribution gap, as well as differences in the timestep boundaries of clips between the large-scale pretraining dataset and the downstream dataset, could be a potential bottleneck that hinders the effectiveness of large-scale pretraining on noisy videos. Strategies aimed at improving robustness to distribution shifts (Sun et al., 2020) represent another promising direction for exploration.

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

# A APPENDIX

In this appendix, we first present additional results on the step classification task (Appendix A.1.1) and procedure planning task (Appendix A.1.2). Then we discuss ablations of our model design, including the choice of attention mechanism (Appendix A.2.1), denoising steps (Appendix A.2.2), classifier-free guidance scales (Appendix A.2.3), and the downstream task performance as we scale up the VEDIT model size (Appendix A.2.4). Furthermore, we provide the PyTorch (Ansel et al., 2024) implementation of VEDIT in Algorithm 1.

## A.1 ADDITIONAL RESULTS

### A.1.1 STEP CLASSIFICATION TASK

In this section, we present additional results on the COIN step classification task, which aims to predict the class labels of single-clip videos. In other words, this task tests only the capability of visual encoders, as the prediction model is not involved. Specifically, for step forecasting and task classification described in the main paper, we design the attentive pooler as a lightweight single cross-attention block with one query token to pool the video clip embedding (e.g., the predicted frame-level `[CLS]` tokens in SigLIP (Zhai et al., 2023)) into a single vector. For the COIN step classification task, we increase the depth of the attentive pooler by adding three additional self-attention blocks before the cross-attention block to aggregate information from the visual features, which we find further improves classification accuracy.

We compare our method, which uses off-the-shelf frozen visual encoders with a trainable attentive classifier, against baseline methods reported in previous works (Lin et al., 2022; Zhong et al., 2023). As shown in Table 7, our method with all five frozen encoders outperforms previous baselines. V-JEPA performs best among the two SSL-based video encoders (i.e., V-JEPA and VideoMAE). Increasing the resolution from 224 to 384 on V-JEPA further boosts accuracy. Additionally, due to the rich information encoded in patch-level tokens, V-JEPA achieves the best performance on the step classification task among all encoders. Moreover, SigLIP, pretrained on both image and text data, outperforms all other encoders except for V-JEPA, demonstrating the effectiveness of using visual-text aligned encoders for procedural activity understanding in instructional videos. However, as the prediction model is not involved, we do not put primary focus on this task in our paper.

| Model | Pretraining Supervision | Pretrain Data | Top-1 Acc. (%) |
|---|---|---|---|
| *Baselines* | | | |
| SlowFast (Feichtenhofer et al., 2019) | Supervised: action labels | Kinetics (Kay et al., 2017) | 32.9 |
| TimeSformer (Bertasius et al., 2021) | Supervised: action labels | Kinetics (Kay et al., 2017) | 48.3 |
| ClipBERT (Lei et al., 2021) | Supervised: captions | COCO+VG (Chen et al., 2015; Krishna et al., 2017) | 30.8 |
| VideoCLIP (Xu et al., 2021) | Unsupervised: ASR | HT100M (Miech et al., 2019) | 39.4 |
| TimeSformer (Bertasius et al., 2021) | Unsupervised: ASR w. MIL-NCE | HT100M (Miech et al., 2019) | 46.5 |
| CLIP (Radford et al., 2021) | Unsupervised: captions | CLIP400M (Radford et al., 2021) | 45.9 |
| DistantSup (Lin et al., 2022) | Unsupervised: ASR + wikiHow | HT100M (Miech et al., 2019) | 54.1 |
| ProceduralVRL (Zhong et al., 2023) | Unsupervised: ASR | HT100M (Miech et al., 2019) | 56.9 |
| *Ours (Frozen encoder w/ lightweight trainable attentive classifier)* | | | |
| DINOv2 (Oquab et al., 2024) | Self-supervised | LVD-142M (Oquab et al., 2024) | 57.9 |
| V-JEPA@224 (Bardes et al., 2024) | Self-supervised | VideoMix2M (Bardes et al., 2024) | 61.4 |
| V-JEPA@384 (Bardes et al., 2024) | Self-supervised | VideoMix2M (Bardes et al., 2024) | **62.7** |
| VideoMAE (Tong et al., 2022) | Self-supervised | Kinetics400 (Kay et al., 2017) | 58.5 |
| SigLIP (Zhai et al., 2023) | Image+Text Pairs | WebLI (Chen et al., 2023) | 61.8 |

Table 7: **Step classification on COIN dataset.** We **bold** and underline the best and the second best models in each task respectively. Our strategy of using strong frozen visual encoder with trainable attentive classifier outperforms all baseline methods.

### A.1.2 PROCEDURE PLANNING TASK

In addition to the main results presented in Table 2, we show in Fig. 3 the comparison of our VEDIT and the vanilla transformer model in (Niu et al., 2024) as we increase the number of transformer blocks. We report success rate (SR), mean accuracy (mAcc), and mean IoU (mIoU) as evaluation metrics on the NIV, COIN, and CrossTask datasets. For a fair comparison, we keep all hyper-parameters the same, with the only change being the number of blocks. We observe that our VEDIT exhibits significantly better stability compared to the vanilla transformer blocks as we scale up the model size, without overfitting to the training set. This finding is consistent with our results in Fig. 7.

## A.2 VEDIT MODEL ABLATIONS

### A.2.1 CHOICE OF ATTENTION MECHANISM

In Fig. 4, we illustrate the differences between our default joint attention in each VEDIT block and self-attention and cross-attention. We denote the observed and unseen video clip embeddings as $v^s$ or $v^t$. In self-attention, we concatenate $v^s$ or $v^t$ along the sequence dimension as a single input to the attention module. In contrast, cross-attention does not utilize self-attention within $v^s$ or $v^t$. Here we conduct ablation study of these attention mechanisms with a prediction model of 3 VEDIT blocks. As shown in Fig. 5, our joint attention outperforms self-attention in step forecasting (50.3 vs. 49.7) and task classification (94.4 vs. 94.3) on the COIN dataset. This proves the usefulness of processing $v^s$ and $v^t$ differently through adaptive normalization layers before inputing to the attention module. In addition, due to the absence of self-attention within $v^s$ or $v^t$, cross-attention performs the worst.

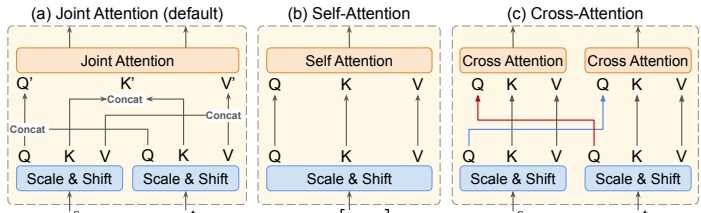

| Top-1 Acc. (%) | (a) | (b) | (c) |
|---|---|---|---|
| Step Forecasting | **50.3** | 49.7 | 47.2 |
| Task Classification | **94.4** | 94.3 | 93.8 |

Figure 5: Top-1 classification accuracy on COIN dataset with different attention mechanisms.

Figure 4: **Ablation of attention mechanisms**, including our default joint attention, self-attention, and cross-attention. We denote seen and target video clip embeddings as $v^s$ and $v^t$ respectively.

### A.2.2 CHOICE OF DENOISING STEPS

Previous work on masked token prediction, such as BERT (Kenton & Toutanova, 2019) for language and MAE (He et al., 2022) for images, can be considered single-step denoising, while diffusion models typically perform single-step denoising during training and multi-step denoising during inference. In this context, we conduct an ablation study on different denoising steps using diffusion timestamps sampled from the Flow Matching Euler Discrete Scheduler (Esser et al., 2024) in our VEDIT training on the COIN step forecasting task. The ablation study here is conducted with a prediction model of 3 VEDIT blocks, and we report the top-1 classification accuracy averaged over three independent runs in Fig. 6. Our results show that applying 20 to 40 denoising steps achieves better accuracy compared to single-step denoising (51.01 for 36 denoising steps v.s. 50.77 for single denoising step). This multi-step denoising allows us to reuse the same VEDIT architecture, with observed and target embeddings scaled and shifted by adaptive normalization layers at different timestamps, without drastically increasing the model's trainable parameters. Additionally, we observe that timestamps that are too sparse (*e.g.*, denoising steps of 4) or too dense (*e.g.*, denoising steps greater than 44) make the model difficult to optimize. We default to use 24 denoising steps in our main paper as it achieves a good balance between computational cost and accuracy.

### A.2.3 CHOICE OF CLASSIFIER-FREE GUIDANCE (CFG) SCALE

We conduct experiments to compare the effect of different CFG scales on the COIN step forecasting and task classification tasks. Specifically, we performed this experiment using the VEDIT architecture with 12 layers and a hidden dimension of 1280, reporting the same top-1 classification accuracy metric used in our main paper.

As shown in the table below, CFG=7 achieves the best step forecasting accuracy, while CFG=3 achieves the best task classification accuracy. Additionally, the accuracy differences across the various CFG scales are relatively small. This is because our model effectively learns parameters that best fit the given CFG scale. Therefore, in the experiments presented in our main paper, we did not fine-tune the CFG parameter and instead used the default CFG value of 7, as in Stable Diffusion 3.

| CFG Scale | 0 | 3 | 5 | 7 | 9 | 11 |
|---|---|---|---|---|---|---|
| Step Forecasting Acc. (%) | 50.73 | 50.43 | 51.30 | 51.85 | 51.74 | 51.28 |
| Task Classification Acc. (%) | 94.70 | 94.84 | 94.71 | 94.67 | 94.47 | 94.42 |

Table 8: Ablation of different classifier-free guidance (CFG) scales.

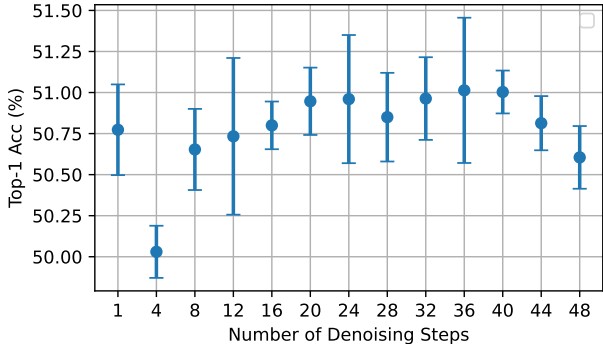

Figure 6: **Ablation of denoising steps.** We report the top-1 accuracy on COIN step forecasting task. A Denoising steps of 24 achieves a good balance between computational cost and accuracy. The numbers are averaged over 3 independent runs.

### A.2.4 SCALABILITY OF VEDIT

Table 9 presents the model architecture details for 10 different scale models we implemented in our scalability results shown in Fig. 7. Specifically, we examined two different sets of hidden dimensions (i.e., 1280 and 2048), with varying numbers of VEDIT blocks ranging from 1 to 18. These parameter settings effectively cover model parameters from 62M to 1.77B (up to ×28 larger in scale). We evaluated these models on the COIN step forecasting task, and the results are averaged over 5 runs. As shown in Fig. 7, with the same number of training epochs, a larger VEDIT model achieves a lower top-1 validation error compared to smaller VEDIT models. This demonstrates that VEDIT is scalable as we increase the model size.

| Model | # Train Params | Layers | Hidden Dim. | # Attn. Heads | Head Dim. |
|---|---|---|---|---|---|
| Hidden Dim. = 1280 | | | | | |
| VEDIT-Single | 62M | 1 | 1280 | 20 | 64 |
| VEDIT-Tiny | 165M | 3 | 1280 | 20 | 64 |
| VEDIT-Small | 342M | 6 | 1280 | 20 | 64 |
| VEDIT-Large | 696M | 12 | 1280 | 20 | 64 |
| VEDIT-XL | 1.05B | 18 | 1280 | 20 | 64 |
| Hidden Dim. = 2048 | | | | | |
| VEDIT-Single | 132M | 1 | 2048 | 32 | 64 |
| VEDIT-Tiny | 418M | 3 | 2048 | 32 | 64 |
| VEDIT-Small | 871M | 6 | 2048 | 32 | 64 |
| VEDIT-Medium | 1.34B | 9 | 2048 | 32 | 64 |
| VEDIT-Large | 1.77B | 12 | 2048 | 32 | 64 |

Table 9: **Details of VEDIT models.** We introduce models of different number of transformer blocks (*i.e.*, layers) with two hidden dimension settings.

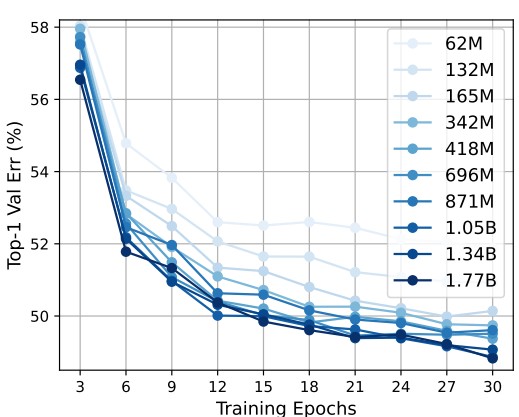

Figure 7: Ablation of different VEDIT model sizes. We report the top-1 validation error on COIN step forecasting task. Our VEDIT demonstrates good scalability as we scale up the model size *up to 28 times from 62M to 1.77B*. The numbers are averaged over 5 independent runs.

**Algorithm 1** Simplified PyTorch Implementation for Each **VEDiT** Block

```python
import torch
from torch import nn

class VEDiT(nn.Module):

    def __init__(
            self, dim, num_attention_heads, attention_head_dim, max_len):

        # adaptive layernorm
        self.norm1_seen = AdaLayerNormZero(dim)
        self.norm1_target = AdaLayerNormZero(dim)

        # normalization layers
        self.norm2_seen = nn.LayerNorm(dim, elementwise_affine=False)
        self.norm2_target = nn.LayerNorm(dim, elementwise_affine=False)

        # FFN
        self.ff_seen = FeedForward(dim=dim, dim_out=dim)
        self.ff_target = FeedForward(dim=dim, dim_out=dim)

        # joint attention
        self.attn = JointAttention(
            dim, num_attention_heads, attention_head_dim, max_len)

    def forward(self, target_emb, seen_emb, temb, target_mask):

        # 1. scale & shift
        norm_target, t_gate_msa, t_shift_mlp, t_scale_mlp, t_gate_mlp = \
            self.norm1_target(target_emb, emb=temb)
        norm_seen, s_gate_msa, s_shift_mlp, s_scale_mlp, s_gate_mlp = \
            self.norm1_seen(seen_emb, emb=temb)

        # 2. joint attention
        seen_attn_output, target_attn_output = self.attn(
            hidden_states=norm_target,
            encoder_hidden_states=norm_seen,
            target_mask=target_mask)

        # 3.1. target: gate, scale & shift
        target_emb = target_emb + t_gate_msa * target_attn_output
        norm_target_emb = self.norm2_target(target_emb)
        norm_target_emb = norm_target_emb * (1 + t_scale_nlp) + t_shift_nlp

        # 3.2. target: feed foreward
        ff_target_output = self.ff_target(norm_target_emb)
        target_emb = target_emb + t_gate_nlp * ff_target_output

        # 4.1. seen: gate, scale & shift
        seen_emb = seen_emb + s_gate_msa * seen_attn_output
        norm_seen_emb = self.norm2_seen(seen_emb)
        norm_seen_emb = norm_seen_emb * (1 + s_scale_nlp) + s_shift_nlp

        # 4.2. seen: feed foreward
        ff_seen_output = self.ff_seen(norm_seen_emb)
        seen_emb = seen_emb + s_gate_nlp * ff_seen_output

        return seen_emb, target_emb
```

Simplified PyTorch Implementation for **JointAttention**

```python
import torch
from torch import nn
import torch.nn.functional as F

class JointAttention(nn.Module):

    def __init__(
        self, dim, num_attn_heads, attn_head_dim, max_len):

        self.heads, self.head_dim = num_attn_heads, attn_head_dim
        self.inner_dim = num_attn_heads * attn_head_dim
        self.max_len = max_len # max number of clips in procedural video

        self.to_q = nn.Linear(dim, self.inner_dim, bias=True)
        self.to_k = nn.Linear(dim, self.inner_dim, bias=True)
        self.to_v = nn.Linear(dim, self.inner_dim, bias=True)

        self.add_q = nn.Linear(dim, self.inner_dim, bias=True)
        self.add_k = nn.Linear(dim, self.inner_dim, bias=True)
        self.add_v = nn.Linear(dim, self.inner_dim, bias=True)

        self.to_out = nn.Linear(self.inner_dim, dim, bias=True)
        self.add_out = nn.Linear(self.inner_dim, dim, bias=True)

        self.rotary_emb = RotaryEmbedding(dim=dim) # rope

    def forward(self, hidden_states, encoder_hidden_states, target_mask):

        residual = hidden_states
        bs = hidden_states.shape[0]

        # 1. concat q, k, and v from projected embeddings
        query = torch.cat([self.to_q(hidden_states),
            self.add_q(encoder_hidden_states)], dim=1)
        key = torch.cat([self.to_k(hidden_states),
            self.add_k(encoder_hidden_states)], dim=1)
        value = torch.cat([self.to_v(hidden_states),
            self.add_v(encoder_hidden_states)], dim=1)

        # 2. get positional indices of seen and target embeddings
        indices = torch.arange(0, self.max_len).repeat([bs, 1])
        seen_pos = indices[~target_mask].reshape([bs, -1])
        target_pos = indices[target_mask].reshape([bs, -1])
        input_pos = torch.concat([target_pos, seen_pos], axis=1)

        # 3. apply rope to query and key
        query = self.rotary_emb.rotate(query, input_pos)
        key = self.rotary_emb.rotate(key, input_pos)

        # 4. apply attention
        hidden_states = F.scaled_dot_product_attention(query, key, value)
        hidden_states = hidden_states.reshape(bs, -1, self.inner_dim)
        hidden_states, encoder_hidden_states = (
            hidden_states[:, : residual.shape[1]],
            hidden_states[:, residual.shape[1] :])

        # 5. linear projection
        hidden_states = self.to_out(hidden_states)
        encoder_hidden_states = self.add_out(encoder_hidden_states)

        return hidden_states, encoder_hidden_states
```

