# OpenReview forum: "VEDIT: Latent Prediction Architecture For Procedural Video Representation Learning"
_ICLR.cc/2025/Conference — ICLR 2025 Poster_

### Official Review · Reviewer_2SCK · 2024-11-03

**Soundness:** 3
**Presentation:** 3
**Contribution:** 3
**Rating:** 6
**Confidence:** 4

**Summary:**

This paper proposes a procedural video representation learning framework that utilizes diffusion transformers to predict visual representation in embedding space. This framework can be effectively transferred to various downstream tasks.

**Strengths:**

1. This paper is well-motivated and easy to follow.

2. VEDiT does not require pre-training the representation space but consumes multiple clips by using pre-trained representation, which is more efficient and applicable, as better representation can improve VEDiT further.

3. Extensive experimental results on various downstream tasks demonstrate the effectiveness of the proposed method.

**Weaknesses:**

1. Question about fair comparison: Without pre-training, I think VEDiT seems to be a large header for the downstream task. I cannot find how the baseline TimeSFormer is evaluated, but if it is evaluated with a simple linear classifier, i.e., linear probing, it seems that VEDiT's performance gain is just from "training more parameters on the pre-trained representation".

2. Typo on page 2 ("fall back on on text annotations" -> "fall back on text annotations")

**Questions:**

Please answer the Weakness part.

---

> ### Author Response · Authors · 2024-11-22
> **Author Response to Reviewer 2SCK**
>
> We would like to sincerely thank the Reviewer for the feedback.
>
> > ### W1: VEDiT with a simple linear classifier
>
> Thank you very much for this valuable question! Firstly, we would like to emphasize that in our main paper, we evaluated our model on five tasks across three benchmark experiments:
> - (1) step forecasting and task classification tasks on COIN dataset
> - (2) procedure planning task on NIV, COIN, and CrossTask
> - (3) long-horizon action anticipation task on Ego4D.
>
> For both (2) and (3), we strictly followed the same experimental settings as the baselines, with the only difference being the replacement of their backbone model with our VEDiT, which has a comparable number of trainable parameters. Therefore, the concern regarding VEDiT using more parameters, as mentioned in this question, is not applicable.
>
> For (1), we focused on exploring the limits of what our model can achieve by comparing it with strong baselines in our main paper. Following the Reviewer’s suggestion, we conducted a complementary analysis using TimeSformer as the backbone while replacing the attentive classifier with the same simple linear classifier used in ProceduralVRL. As shown in the table below, using a simple linear classifier still outperforms the previous SoTA (ProceduralVRL). Additionally, we would like to emphasize that this comparison is not strictly apple-to-apple, as VEDiT removes the need for any HowTo100M pretraining or text supervision.
>
>
> ||||
> |-|-|-|
> | | Step Forecasting Accuracy (%) | Task Classification Accuracy (%) |
> | ProceduralVRL (previous SoTA) | 46.8 | 90.8 |
> | VEDiT with simple linear classifier | 47.38 | 91.29 |
> | VEDiT with attentive classifier | 48.44 | 91.89 |
>
>
>
> > ### W2: Typo
>
> Thanks for pointing this out! We’ve updated our pdf and corrected this typo.
>
>
> - - -
> We would like to once more sincerely thank you for all the comments and very useful feedback. We think that we have addressed in depth all Reviewer's questions. If the Reviewer has any additional questions, please let us know, we would be more than happy to answer them.

---

> ### Author Response · Authors · 2024-11-25
> **Official Comment by Authors**
>
> Dear Reviewer 2SCK,
>
> We would like to once more sincerely thank you for your detailed feedback and apologize for taking your time. We believe that we have addressed all your questions, in particular: (1) quantitative results of VEDiT with single linear classifier, as well as highlights of our fair-comparison results on procedural planning and long-horizon action anticipation tasks, and (2) correction of the typo mentioned by the Reviewer.
>
> We would appreciate your response soon as the deadline for the discussion period ends tomorrow. If our rebuttal is adequate to your questions, we would also appreciate an update in your evaluations. Once more, thank you very much for your reviews and feedback!
>
> Yours sincerely,
>
> The Authors

---

> > ### Comment · Reviewer_2SCK · 2024-11-25
> >
> > Thanks for the detailed response. The rebuttal has addressed my concerns. I will keep my current positive score and increase my confidence.

---

> > > ### Author Response · Authors · 2024-11-26
> > >
> > > Thanks for your positive feedback! We are very glad to hear that your concerns have been well-addressed!
> > >
> > > Once more, we sincerely appreciate the time and effort you devoted to reviewing our paper, and we truly appreciate all your valuable comments and suggestions!

---

### Official Review · Reviewer_czP8 · 2024-11-04

**Soundness:** 2
**Presentation:** 2
**Contribution:** 2
**Rating:** 6
**Confidence:** 4

**Summary:**

This work questioned the necessity and effectiveness of pretraining by develop a SoTA method for forecasting and procedural planning without the need for pretraining the prediction model. The proposed model, VEDIT, leverage diffusion transformer and attentive classfier directly trained for downstream tasks. Empirical results showed that VEDIT outperform all baselines.

**Strengths:**

1. Nice presentation and well-designed figures.

2. Strong empirical performance: VEDIT outperform all baselines quantitatively on all three benchmarks.

**Weaknesses:**

1. Definition of pre-training: The most confusing point for me is that the authors claim the results of this work could challenge the necessity of pretraining. However, first of all, VEDiT itself uses a pretrained encoder. How is this different from what the authors define as pretraining? Secondly, isn’t the training of the diffusion part considered pretraining? I think the authors should make a stricter distinction and definition of pretraining to help readers clearly understand what they really want to refer to by "pretraining."

2. Training objective of DiT: If the training of the diffusion part isn’t considered pretraining, then what is the role of diffusion here? The authors should use mathematical formulas to clarify exactly what their loss function is; the current explanation is too vague. Additionally, starting from line 233, the description of diffusion training seems to suggest a possible misunderstanding of diffusion training.

3. Test-time training [1]:  This is just a suggestion, but based on the authors' model, I think that applying test-time training exclusively to the diffusion component could further enhance the model's performance during inference.





[1]. Sun, Y., Wang, X., Liu, Z., Miller, J., Efros, A. and Hardt, M., 2020, November. Test-time training with self-supervision for generalization under distribution shifts. In International conference on machine learning (pp. 9229-9248). PMLR.

**Questions:**

All of my questions are listed in the weakness part.

---

> ### Author Response · Authors · 2024-11-22
> **Author Response to Reviewer czP8**
>
> We would like to sincerely thank the Reviewer for the valuable feedback and revising suggestions.
>
> > ### W1: Definition of pre-training
>
> Thank you very much for the valuable comment. We have revised our PDF to clarify the definitions of the two types of “pretraining” mentioned in our paper. Specifically, we now distinguish between **“pretraining on single video clips”** and **“pretraining on video clip sequences”** in the updated PDF. When we state that our model does not require large-scale pretraining, we are specifically referring to the lack of necessity for “pretraining on video clip sequences.”
>
> As mentioned in **L043–L049**, existing procedural video representation learning approaches typically extend the traditional single-clip pretraining strategy (used in activity understanding for short video clips) to large-scale video clip sequences, such as HowTo100M, by pretraining both the video encoder and prediction model. In contrast, our findings reveal that pretraining the prediction model on such large-scale and noisy video clip sequences does not significantly improve performance on downstream tasks. By re-using a frozen video encoder pretrained on single video clips and directly training the prediction model on downstream datasets, we achieve state-of-the-art (SoTA) results on several evaluation benchmarks, as outlined in the experiment section.
>
>
>
> > ### W2: Training objective of VEDiT
>
> Thank you very much for the comment. We have added a new paragraph “Training Objective of VEDiT” in the Methods section of our revised PDF to explicitly explain the training objective of our VEDiT. As illustrated in the right part of **Fig. 1** in our main paper, rather than using a diffusion loss, the training objective of our pipeline is a cross-entropy loss, which aligns with downstream classification tasks. Instead, we utilize flow-matching/diffusion not as a loss function but as a mechanism to schedule the T timestamps, iteratively denoising random Gaussian noise into predicted video embeddings.
>
>
> Such training objective design is one of the key innovations and distinctions of our method, compared with previous approaches to video embedding prediction for downstream tasks [1, 2]. Previous works typically rely on a combination of multiple loss functions (such as video embedding reconstruction loss and video-language matching loss) for supervision, which enforces alignment of the predicted video embeddings with often noisily annotated language descriptions. In contrast, our usage of a single cross-entropy loss allows the optimization target to align more effectively with downstream datasets.
>
> Thanks again for proposing this important question. We believe that the illustration here as well as the revised method section in our updated PDF can better help the readers better understand our training objective.
>
> [1] Zhong, Yiwu, et al. "Learning procedure-aware video representation from instructional videos and their narrations.", CVPR 2023
>
> [2] Lin, Xudong, et al. "Learning to recognize procedural activities with distant supervision.", CVPR 2022
>
>
> > ### W3: Test-time training
>
> Thank you very much for the pointers! Indeed, we agree that such test-time training approach could potentially benefits the generalization ability of our VEDiT, especially for the scenario when we do pretraining on video clip sequences sampled from HowTo100M and then do inference on downstream datasets such as COIN.
>
> A high-level idea of adopting such test-time training is to encode the video clips into some global-level visual embeddings, then give as input similar as the diffusion timestamp (i.e., the red area in Fig. 2 of our main paper)  to the VEDiT architecture through adaptive normalization layers. In this way, we can allow the predicted target clip embeddings of a video to be transformed according to the data distribution of this video. We have included this paper in Sec. 6 of our updated PDF as a potential strategy to improve robustness to distribution shifts, highlighting it as a promising future direction to explore.
>
>
>
> - - -
> We would like to once more sincerely thank you for all the comments and very useful feedback. We think that we have addressed in depth all Reviewer's questions. If the Reviewer has any additional questions, please let us know, we would be more than happy to answer them.

---

> > ### Comment · Reviewer_czP8 · 2024-11-26
> > **reply to rebuttal**
> >
> > I appreciate the authors' clarification. They address my main concerns and questions about this paper. I will raise score to 6.

---

> > > ### Author Response · Authors · 2024-11-26
> > >
> > > Thanks for your positive feedback! We are very glad to hear that your concerns and questions have been well-addressed!
> > >
> > > Once more, we sincerely appreciate the time and effort you devoted to reviewing our paper, and we truly appreciate all your valuable comments and suggestions!

---

> ### Author Response · Authors · 2024-11-25
> **Official Comment by Authors**
>
> Dear Reviewer czP8,
>
> We would like to once more sincerely thank you for your detailed feedback and apologize for taking your time. We believe that we have addressed all your questions, in particular: (1) clarification of the two types of pretraining (i.e., “single-clip pretraining” and “pretraining on clip sequences”) mentioned in our paper (see: updated PDF), (2) inclusion of detailed explanation of the training objective of VEDiT in the method section of our updated PDF (3) discussion about how to incorporate test-time training into our framework.
>
> We would appreciate your response soon as the deadline for the discussion period ends tomorrow. If our rebuttal is adequate to your questions, we would also appreciate an update in your evaluations. Once more, thank you very much for your reviews and feedback!
>
> Yours sincerely,
>
> The Authors

---

### Official Review · Reviewer_ajEB · 2024-11-07

**Soundness:** 3
**Presentation:** 3
**Contribution:** 3
**Rating:** 6
**Confidence:** 4

**Summary:**

This paper deals with  Procedural Video Representational Learning, in which a model predicts steps in a underlying process (eg cooking a dish). The goal is to learn a representation that enables prediction and understanding of the semantics of step-by-step processes through videos, often egocentric or how-to videos. Most works in this space pre-train models on data from HowTo100M with extracted ASR captions which is time-consuming and expensive. This paper's contribution is (1) a novel architecture for the procedural video task based on diffusion / flow matching in latent state-action space and (2) a method that relies only on frozen pre-trained image encoders rather than explicit procedural weakly supervised pretraining. In doing so, they achieve SoTA results.

In general, I think this is a reasonably strong paper, and the only weakness to me is that the architecture has slightly limited novelty; it's not very clear to me exactly how this differs from a standard diffusion transformer. The use of a frozen encoder, while simple, works well and avoids having to do expensive ASR-based pre-training. The models are also scaled to high parameter counts and thus will be useful to the community.

**Strengths:**

1) The proposed architecture (diffusion in latent space between steps) is interesting. I think doing diffusion on the latent embeddings is a smart way to generate new embeddings that can be used for predicting final class embeddings.
2) The results surpass existing SOTA and the ablations are comprehensive. The method also avoids relatively expensive pre-training on direct step data, making use of easily available visual encoders like SigLIP. The appendices are thorough and explain relevant design decisions for the proposed architecture.
3) The models trained are relatively large and will be helpful for the community if made available.

**Weaknesses:**

1) The models trained are quite large, with the largest having 1.77B parameters. Given that the improvement over SOTA is relatively minor in most cases, it's not clear that scaling this method really works - could you comment on why this might be the case?

2) The architectural novelty is a little limited. The idea of doing diffusion to produce intermediate embeddings isn't new (as is stated in the paper) and this paper differs from prior work mostly by using a frozen SigLIP encoder rather than one pre-trained on HowTo100M (if i am understanding correctly).

3) It's not super clear how the proposed VeDIT differs from standard diffusion transformer blocks except for the embedding concatenation - am I missing something? If so, could you explain further?

**Questions:**

1) It's stated the task is either predicting the label of unseen events in between two input events (procedure planning), or predicting the label of a future event.  In the case of procedure planning, how does the model know how many clips to generate?  It's not obvious from the system figure (Fig 2) how multi-clip generation works - does that require multiple runs of the entire diffusion model? Is this number fixed?

2) From an intuition standpoint, why is diffusion the right choice for generating an embedding given a set of input embeddings? Are there no faster alternatives? What was the reasoning behind this choice compared to other alternatives?

---

> ### Author Response · Authors · 2024-11-22
> **Author Response to Reviewer ajEB (Part 1/3)**
>
> We would like to sincerely thank the reviewer for the insightful and detailed comments.
> Responses to weakness 1 and 2 (W1 and W2), as well as question 1 and 2 (Q1 and Q2) are provided below. Weakness 2 (W2) is addressed in the global response “Q1: Summary of contributions and novelty of VEDiT”.
>
> > ### W1: Scalability of VEDiT
>
> Thank you very much for the comment.
>
> - **For the COIN step forecasting and task classification tasks:** As a reference, ProceduralVRL [1] includes in Table 1 of their paper an “upper bound” of their result by assuming an oracle ranking function that always selects the correct prediction from 5 outputs sampled from their model (i.e., Ours oracle-5). Under such strong assumptions, the performance of this oracle-5 is only **51.8**, which is the same as our SigLIP + VEDiT in Table 1 shown in our paper. Therefore, as shown in Fig. 7, we would like to emphasize that the top-1 validation error improvement from 52 to 48.5 as we scale the model from 62M to 1.77B is still non-trivial.
>
> - **For the procedural planning task on COIN, CrossTask and NIV datasets:** In addition, we have shown in Fig. 3 that the default vanilla transformer model used in SCHEMA clearly suffers from severe performance degradation as we increase the number of transformer blocks, while VEDiT doesn’t have such drawbacks. This further proves the scalability of our VEDiT compared with vanilla transformer models on such procedural video tasks.
>
>
> [1] Zhong, Yiwu, et al. "Learning procedure-aware video representation from instructional videos and their narrations.", CVPR 2023.
>
>
> > ### W2: Our novelty
>
> Thank you very much for this question. We have addressed this question in the general comment “Q1: Summary of contributions and novelty of VEDiT”. Please let us know if there are any further concerns about the novelty of VEDiT. We will do our best to address your concerns and provide detailed illustrations as needed.

---

> ### Author Response · Authors · 2024-11-22
> **Author Response to Reviewer ajEB (Part 2/3)**
>
> > ### W3: VEDiT v.s. DiT
>
> Thank you very much for this excellent question. Indeed, our VEDiT shares some similarities with DiT. As mentioned in the global response, “Q1: Summary of Contributions and Novelty of VEDiT,” our primary motivation is to demonstrate that _a simple generative model architecture, paired with a single task-specific loss function and no additional supervision, can achieve SoTA performance._
>
> However, VEDiT differs from DiT in several key aspects of its design, integrating a range of the latest techniques from the visual generation community. These include the DiT-based architecture, rotary position embeddings (RoPE), adaptive normalization layers, joint attention mechanisms, as well as training objectives. Below, we explain in detail how VEDiT distinguishes itself from DiT:
>
>
> - **DiT operates on patches instead of more abstract video embeddings.** DiT takes the image patches as input and reconstruction targets. In contrast, our VEDiT is designed to operate on the more abstract video embeddings (e.g., SigLIP, DINOv2). Such design allows VEDiT to do effective prediction on the high-level and abstract visual embeddings, instead of focusing on low-level visual signals.
>
>
> - **VEDiT is a dual-branch architecture that uses joint attention to fuse the information between observed and target clips** The original DiT is a single-branch architecture with self-attention, while our VEDiT is designed as a dual-branch architecture that uses two sets of adaptive normalization layers to transform the video embeddings of observed and target clips. Our joint attention design not only allows the self-attention among observed/target clips, but also cross-attention between them. Ablation studies on the choice of attention mechanism are shown in Appendix Sec. A.2.1 proves the effectiveness of our attention design.
>
>
> - **The training objectives between VEDiT and DiT are different** This is another major difference with respect to DiT. Our VEDiT is designed to be trained with Cross-Entropy loss that is directly aligned with the downstream tasks while the training objective in DiT is the reconstruction loss between input and output video embeddings. In addition, the flow-matching noise scheduler in VEDiT is used as a tool to sample the multi-step denoising trajectory of noise scales, which is different from the single-step denoising objective used in the training of DiT and other vision generative models.
>
>
> - **DiT is a class-conditional generative model.** DiT takes a (ImageNet) class label as global condition to guide the model to denoise the images from random noise. In contrast, our VEDiT is designed as a vision-embedding-conditioned model, with input as the concatenation of both observed and target clips. Since our default VEDiT design doesn’t need any text supervision (extension to multimodal VEDiT is discussed in the reply “W3: Multimodal Integration” to reviewer ajEB), other than diffusion timesteps, we don’t need global condition in our default model architecture design.
>
>
> - **VEDiT employs RoPE to model the temporal dependencies of clips procedural videos** The original DiT uses frequency-based positional embeddings (sine-cosine version) to encode the _spatial_ information among patches. In contrast, for procedural understanding, encoding of _temporal_ information is needed. Therefore, we apply RoPE on top of the video embeddings output from pretrained visual encoders to enable the model to understand temporal dependencies of video embeddings.
>
>
> > ### Q1: Is the number of clips to generate in procedural planning tasks fixed?
>
> Yes, we follow the settings of previous works on this task, which assumes that the number of clips to generate is fixed. We strictly follow the same experiment settings in the baselines, with the only difference of replacing their Transformer model with our VEDiT with similar trainable parameters. We added clarification about this point in our **updated PDF**.

---

> ### Author Response · Authors · 2024-11-22
> **Author Response to Reviewer ajEB (Part 3/3)**
>
> > ### Q2: Intuition why flow-matching/diffusion works for video embedding prediction
>
> Thank you very much for this great question! Below we first illustrate the high-level intuition why we use diffusion/flow-based noise scheduler + Cross-Entropy loss for procedural tasks, then show the advantages of applying multi-step denoising for video embedding prediction.
>
> **1. Intuition for diffusion/flow-based noise scheduler + Cross-Entropy loss:**
>
> From a high-level intuition, using a diffusion/flow-based noise scheduler with multi-step denoising is just one of the strategies to predict the video embeddings of target clips. A key innovation and distinction of our method, compared with previous approaches to video embedding prediction for downstream tasks [1, 2]—which typically rely on a combination of multiple loss functions (such as video embedding reconstruction loss and video-language matching loss) for supervision—is that our pipeline can be effectively trained with a single Cross-Entropy loss. Unlike previous works that enforce alignment of the predicted video embeddings with often noisily annotated language descriptions, using a single Cross-Entropy loss allows the optimization target to align more effectively with downstream datasets.
>
>
> **2. Intuition about why we adopt multi-step denoising with VEDiT**:
>
> - **2.1 Larger DiTs perform better than smaller DiTs:** One major motivation for us to adopt a deep model (through multi-step denoising) comes from the finding in Diffusion Transformers (DiT). In Fig. 9 of their paper, they find that with the same amount of training compute (measured by Gflops), larger DiT models are more computationally efficient than smaller DiT models. In addition, small DiT models, even when trained longer, eventually underperform larger DiT models trained for fewer steps.
>
> - **2.2 Multi-step denoising is equivalent to creating a deep model, but in a parameter-efficient way without increasing the number of trainable parameters:** One good property of DiTs, compared with the vanilla transformer models, is that it is able to perform embedding prediction by conditioning on different levels of noises / timesteps. To be specific, DiTs leverage adaptive normalization layers to rescale and shift the input embeddings according to the current time step before passing the embeddings into attention modules (see Fig. 2 “scale, shift, gate” for details). In other words, we can utilize the same model architecture, but run iteratively with different time steps to create a large/deep DiT architecture without multiplying/amplifying the number of trainable parameters  .
>
> - **2.3 Multi-step denoising is more robust compared with single-step denoising:** From 1 and 2 mentioned above, we now know that multi-step denoising can create a “deep” DiT architecture that is proved to be more effective than smaller DiTs. In addition, another good property of multi-step denoising is that it is more robust to single-step denoising. Intuitively, VEDiT denoises the target clip video embedding from random gaussian noise (similar as other denoising models). In each intermediate denoising step, it can freely change the predicted embeddings of target clip based on the information from observed clips. In contrast, single-step denoising does not have such flexibility, and is less robust since it only has a single chance to do prediction from the random gaussian noise.
>
>
> **3. Performance boost via multi-step denoising from empirical results**
>
> On the one hand, in Fig. 6 of our appendix, we have shown that adopting multi-step denoising with a number of denoising steps between 16 and 44 can boost performance compared with single step denoising. On the other hand, we would like to kindly point out that even with single-step denoising, our classification accuracy of **50.77** still outperforms **46.8** from the SOTA baseline (ProceduralVRL [1]) on the step forecasting task. As a reference, ProceduralVRL includes in Table 1 of their paper an “upper bound” of their result by assuming an oracle ranking function that always selects the correct prediction from 5 outputs sampled from their model (i.e., Ours oracle-5). Under such strong assumptions, the performance of this oracle-5 is only 51.8. Therefore, we would like to emphasize that the top-1 classification accuracy improvement from previous SoTA (46.8) to our VEDiT with multi-step denoising (51.8) is non-trivial.
>
>
>
> [1] Zhong, Yiwu, et al. "Learning procedure-aware video representation from instructional videos and their narrations.", CVPR 2023
>
> [2] Lin, Xudong, et al. "Learning to recognize procedural activities with distant supervision.", CVPR 2022
>
>
>
>
> - - -
> Once more, we sincerely thank you for all the comments and very useful feedback. We think that we have addressed all the questions in depth. If the reviewer has any additional questions, please let us know, and we will be more than happy to answer them.

---

> ### Author Response · Authors · 2024-11-25
> **Official Comment by Authors**
>
> Dear Reviewer ajEB,
>
> We would like to once more sincerely thank you for your detailed feedback and apologize for taking your time. We believe that we have addressed all your questions, in particular: (1) detailed discussion about the technical novelty and contribution of VEDiT, (2) architecture comparison of VEDiT with the vanilla DiT, (3) intuition about why diffusion/flow-based multi-step denoising works for video embedding prediction, and (4) discussion of the non-trivial scalability of VEDiT.
>
> We would appreciate your response soon as the deadline for the discussion period ends tomorrow. If our rebuttal is adequate to your questions, we would also appreciate an update in your evaluations. Once more, thank you very much for your reviews and feedback!
>
> Yours sincerely,
>
> The Authors

---

> > ### Comment · Reviewer_ajEB · 2024-11-26
> >
> > Thank you for the detailed response.
> >
> > You state that  the improvement from "52 to 48.5 as we scale the model from 62M to 1.77B" is non-trivial. Is it really? GPT-2 at 124M to 1.5B has orders of magnitude difference. This number is pretty small. Is there something about COIN that makes it really hard to do better on this task? Scaling by 28x shouldn't yield a improvement of just 5%, right?
> >
> > I think most of the differences you outlined between VeDiT and DiT are pretty minor and I don't think there is a real case to support any claim of "novel architecture". DiT doesn't directly operate on patches, it operates on latents produced by a tokenizer like VQ-VAE. The only architectural difference is the dual-branch part but that's just concatenation, and doesn't really merit being one of the main contributions of the paper. The only real difference is the training objective. I don't think this is necessarily a bad thing, but I think it might be helpful to refocus the writing of the paper on what is actually the contribution.
> >
> > You also don't really address the speed issue. Multi step denoising is very slow and there is a real tradeoff to doing this. Why does the performance boost merit significantly slower runtime?
> >
> > I will keep my rating at a 6. The reason is that while I think this paper is well-executed, the scope is rather narrow (focused only on a single task, procedural video forecasting) and thus limits its impact. That being said, it does have research value and deserves to be accepted.

---

> > > ### Author Response · Authors · 2024-11-26
> > >
> > > Dear Reviewer ajEB,
> > >
> > > Thank you for your positive feedback! We sincerely appreciate the time and effort you devoted to reviewing our paper and providing valuable comments and suggestions.
> > > We are delighted to hear that you find our work to be of _good research value, well-executed, and deserving of acceptance_. Many thanks!
> > >
> > > Below, we would like to kindly address your latest comments:
> > >
> > > **Performance Improvement on the COIN Dataset**
> > >
> > > We acknowledge that the score improvement observed after scaling the VEDiT model size is not as significant as expected on the COIN dataset.
> > > - As noted in our previous response, one reason for this is that the oracle "upper bound" proposed in ProceduralVRL—which assumes a perfect ranking function that always selects the correct prediction from 5 outputs sampled from the model—achieves a score of only 51.8. This is actually the same as the best score achieved by the 1.7B VEDiT model with multi-step denoising. So getting scores higher than this might be very hard on COIN dataset.
> > > - Additionally, we agree that the COIN dataset, along with other widely-used procedural video understanding datasets (e.g., CrossTask, NIV), has its own limitations. Developing a better evaluation benchmark for vision-based procedural video understanding with high-quality, human-annotated labels is a non-trivial job. This endeavor, parallel to the contributions of our paper, is highly valuable to the research community and deserves further exploration.
> > >
> > > **Refocusing the Paper’s Contributions**
> > >
> > > Thank you very much for this insightful suggestion!
> > >
> > > - As outlined in “Q1: Summary of Contributions and Novelty of VEDiT” in our general response, one of the primary motivations of our paper is to bridge the gap between video generation and understanding using a diffusion-based architecture. Accordingly, VEDiT was intentionally designed to inherit from DiT, but with important and meaningful modifications that makes it still distinguished from the original DiT design. At a high level, similar to the use of MLLM as a unified architecture for both understanding and generation, our goal here is to design a diffusion-based architecture capable of achieving this unification.
> > >
> > > - As you suggested, we will also place greater emphasis on the contribution and novelty of our training objective, which—based on our understanding—is thoroughly studied for the first time in this paper. We believe this aspect could significantly benefit future research in this direction.
> > >
> > > **Speed Concerns**
> > >
> > > Thank you for highlighting this issue!
> > >
> > > - As mentioned in “Q2: Computational Cost of VEDiT” in our general response, the speed of multi-step denoising, even with 24 steps, is only 1.32x slower for inference and 2.02x slower for training compared to single-step denoising. If the user is very sensitive to the training/inference time, they can just use single-step denoising, which can still achieve better performance compared with previous SoTAs.
> > > - Regarding the speed-performance tradeoff, we view this as an empirical question, with the optimal choice depending on the user and specific downstream applications. Furthermore, exploring improvements through techniques such as model distillation, quantization, and hardware-level optimization could be a valuable direction for future work. However, these optimizations fall outside the scope of this paper and are left for future exploration.
> > >
> > > ---
> > > Finally, we would like to reiterate our sincere gratitude for your detailed comments and valuable feedback. Your suggestions are incredibly helpful in improving our manuscript. We will follow your recommendations for refining the paper's key contributions and strengthen it further in the camera-ready version.

---

### Official Review · Reviewer_Xzeh · 2024-11-09

**Soundness:** 2
**Presentation:** 2
**Contribution:** 2
**Rating:** 6
**Confidence:** 4

**Summary:**

This paper aims to advance video representation learning by proposing a diffusion transformer specifically for procedural video representation. It demonstrates that a frozen, pretrained visual encoder paired with a prediction head can achieve promising performance across five procedural learning tasks.

**Strengths:**

1. The proposed method avoids the need for compute-intensive pretraining of the prediction model. It uses a pretrained, frozen visual encoder to capture robust visual representations, eliminating the need for additional supervision (e.g., language or ASR) commonly required in prior methods

2.The model's empirical evaluation demonstrates significant performance improvements on procedural video benchmarks like COIN, CrossTask, and Ego4D, showing gains in task classification accuracy and long-horizon action anticipation.

**Weaknesses:**

1.Lack of Sufficient Innovation: The VEDIT model's components lack distinct novelty. (1) VEDIT’s architecture is primarily a combination of existing techniques—pretrained visual encoders and diffusion transformers—rather than introducing new modules. (2) The method’s adaptation of diffusion transformers to work in the latent embedding space of video representations has been explored previously, as in PPDP.

2.Omission of Computational Cost Analysis: The paper does not address the computational cost of the method. Due to the iterative nature of the diffusion process, VEDIT incurs slower inference times than models that make predictions in a single forward pass. This characteristic may limit its use in real-time or low-latency applications.

3.Lack Disccusion of Multimodal Integration: By not incorporating textual data or language models, the method overlooks semantic information that could enrich the understanding of procedural steps, especially in instructional videos that include narration.

**Questions:**

See details in 'Weakness' section.

---

> ### Author Response · Authors · 2024-11-22
> **Author Response to Reviewer Xzeh**
>
> We sincerely thank the Reviewer for their insightful and detailed comments. Regarding the questions raised in Weaknesses 1 and 2 (W1 and W2), we have addressed them in the general response. For Weakness 3 (W3), we provide our answer below.
>
> > ### W1 & W2: Novelty and Computational Cost of VEDiT
>
> Thank you very much for this question. We have addressed this question in the general comments “Q1: Summary of contributions and novelty of VEDiT” and “Q2: Computational cost of VEDiT. Please let us know if there are any further concerns about the novelty or the computational cost of VEDiT. We will do our best to address your concerns and provide detailed illustrations as needed.
>
>
>
> > ### W3: Multimodal Integration
>
> Thank you very much for this excellent comment! In the response below, we first emphasize our goal of designing a vision-centric model and then present additional experiments demonstrating how multimodal integration (e.g., text) can be achieved with our architecture.
>
> **VEDiT is designed as a vision-centric model without text supervision**
>
> As mentioned in L148–L150 of our main paper, VEDiT is designed to be a simple yet effective vision-centric model architecture. Compared to SoTA models for procedural video representation learning (e.g., ProceduralVRL, DistantSup), which rely on extra step-level or video-level text descriptions, we would like to emphasize that VEDiT achieves SoTA performance while eliminating the need for any HowTo100M pretraining or text supervision.
>
>
> **Additional preliminary experiments on multimodal (text) integration into VEDiT**
>
> Following the Reviewer’s comment, we conducted preliminary experiments exploring multimodal integration. Specifically, we utilized the video captioning model CogVLM2-Caption [1] to generate dual-level captions: **(1) video-level captions** summarizing the entire video, and **(2) step-level captions** for each clip in the COIN video. We then used the T5-XXL text encoder to convert these captions into text embeddings. The figure illustrating the overall model architecture is contained in [this anonymous github repo](https://anonymous.4open.science/r/VEDiT_ICLR2025_Rebuttal-7A49).
>
> - For (1), we employed an attentive pooler with three transformer blocks and a single query token to aggregate word-level text embeddings into a single global text embedding. This video-level global text embedding was added to the timestep embedding after linear projection. Intuitively, this global-level text influences all visual embeddings through scale, gate, and shift operations in the adaptive normalization layers.
>
> - For (2), we used an attentive pooler with three transformer blocks and 16 query tokens to aggregate word-level text embeddings into fewer tokens for computational efficiency. These clip-level text embeddings were processed similarly to the video embeddings. Specifically, we added a separate branch to the VEDiT architecture with independent adaptive normalization layers, enabling better alignment between text and visual embeddings. The Q, K, and V matrices were updated to include a concatenation of observed video embeddings, predicted video embeddings, and text embeddings.
>
> We evaluated this setup on the COIN step forecasting task. As shown in the table below, adding (2) step-level captions resulted in a top-1 classification accuracy of **51.2**, outperforming the vision-only model accuracy of **50.6**. However, incorporating (1) global-level captions negatively impacted classification accuracy. We hypothesize this could be because video-level captions struggle to encode sufficient information to guide next-step predictions effectively.
>
> Finally, we would like to acknowledge that this exploration of multimodal integration is preliminary, and the current implementation and design choices are not fully optimized. Further refinement in this area, along with the effective combination of VEDiT with large language models (LLMs), represents a promising direction for future research.
>
>
> |||
> |-|-|
> | | Step Forecasting Accuracy (%) |
> | vision-only (default VEDiT) | 50.6 |
> | vision + (1) global-level captions | **51.2** |
> | vision + (2) step-level captions | 48.0 |
> | vision + (1) global-level captions + (2) step-level captions | 48.5 |
>
>
>
>
> [1] Yang, Zhuoyi, et al. "Cogvideox: Text-to-video diffusion models with an expert transformer." arXiv 2024.
>
>
>
> - - -
> We would like to once more sincerely thank you for all the comments and very useful feedback. We think that we have addressed in depth all Reviewer's questions. If the Reviewer has any additional questions, please let us know, we would be more than happy to answer them.

---

> ### Author Response · Authors · 2024-11-25
> **Official Comment by Authors**
>
> Dear Reviewer Xzeh,
>
> We would like to once more sincerely thank you for your detailed feedback and apologize for taking your time. We believe that we have addressed all your questions, in particular: (1) detailed discussion about the technical novelty and contribution of VEDiT, (2) computational cost of VEDiT, and (3) new experiments on multimodal integration of VEDiT with languages.
>
> We would appreciate your response soon as the deadline for the discussion period ends tomorrow. If our rebuttal is adequate to your questions, we would also appreciate an update in your evaluations. Once more, thank you very much for your reviews and feedback!
>
> Yours sincerely,
>
> The Authors

---

> ### Author Response · Authors · 2024-12-01
>
> Dear Reviewer Xzeh,
>
> Hope you had a wonderful Thanksgiving holiday week! We would like to sincerely thank you once again for your detailed feedback and apologize for taking up your time. We believe that we have addressed all your questions, particularly: (1) a detailed discussion about the technical novelty and contributions of VEDiT, including a thorough comparison with PPDP, (2) computational cost of VEDiT, and (3) new experiments on the multimodal integration of VEDiT with language.
>
> As the deadline for the extended discussion period ends tomorrow, we would greatly appreciate your response soon. If our rebuttal adequately addresses your questions, we kindly request an update to your evaluations. Once again, thank you very much for your reviews and feedback!
>
> Yours sincerely,
>
> The Authors

---

### Official Review · Reviewer_yPQ6 · 2024-11-10

**Soundness:** 3
**Presentation:** 3
**Contribution:** 2
**Rating:** 6
**Confidence:** 3

**Summary:**

This paper introduces a novel pipeline,VEDIT, for procedural classifciation task using diffusion transformers without large-scale pre-training or additional language-based supervision. VEDIT leverages a diffusion transformer-based model to operate on latent embeddings derived from frozen, pretrained visual encoders. It is designed to predict the future steps in multi-step tasks (e.g., cooking, assembly) by iteratively denoising embeddings in the latent space, which are generated from observed steps. Authors conducted adequate experiments to demonstate that VEDIT achieves state-of-the-art (SoTA) results on various tasks like step forecasting, task classification, and procedure planning across multiple datasets (e.g., COIN, Ego4D-v2, CrossTask, and NIV).

**Strengths:**

Generally, the primary strength of this paper is the introduction of the novel diffusion-based representation learning method into the video understanding field. VEDIT achieves significant performance gains across multiple procedural tasks, such as step forecasting and task classification.

**Clarity**: Most of this paper is well-written, with clear explanations and a solid grounding in relevant literature.

**Originality**: The main idea of training the VEDiT using the CrossEntropyLoss from the downstream task for video representation learning is novel and has the potential to be applied to various temporal grounding tasks beyond classification.

**Versatility**: VEDIT stands out for its versatility across tasks. VEDIT successfully handles multiple procedural learning tasks, including step forecasting, task classification, and procedure planning, proving its robustness in diverse scenarios.

**Quality**: The overall quality of the paper is high. The empirical results are well-supported by ablation studies, which explore the impact of different model components, and the paper presents a thorough comparison against state-of-the-art methods, showcasing the strengths of the proposed model.

**Weaknesses:**

There are some major concerns:
- While the approach of training VEDiT using CrossEntropyLoss from downstream tasks is novel, it remains challenging for the audience to understand why the diffusion model can learn with this design and the training pipeline. From the destination in L239-L241 and  L301-L303,  it seems the sampled Gaussian noise will be passed to $12 \times 24= 288 $ transformer blocks in a single forward pass, where $12$ is the number of transform blocks and $24$ is the denoising steps. It seems that the computing needs are much higher than baseline methods, and it may not support other inference parameters, including denoising steps and classifier-free guidance scales, since they are fixed in training. It isn't clear how well it works when reducing the sampling steps to lower steps to balance the computing cost and performance boost.
- Using diffusion models for video embedding prediction is time-consuming since it requires $N$ times longer inference time. However, authors didn't discuss the limitation in this paper.


In conclusion, these are concerns and weaknesses that could further improve the paper. However, the weaknesses outweigh the strengths. Consequently, I am leaning towards suggesting rejection of this paper. I am looking forward to the authors' rebuttal and clarification.

**Questions:**

I also have a variety of questions that did not significantly impact the rating for the paper:
- The paper demonstrated the advantage of using VEDiT on chunk-level classification tasks. Can the approach be extended to other temporal understanding tasks that predict start/end time? e.g., action grounding
- The paper entirely leverages flow-matching sampling with a fixed number of sampling steps. How's the model's performance with different scheduler/sampling parameters?

**Details Of Ethics Concerns:**

No ethics concerns.

---

> ### Author Response · Authors · 2024-11-22
> **Author Response to Reviewer yPQ6 (Part 1/4)**
>
> We would like to sincerely thank the Reviewer for the feedback. In the rebuttal below, we use “W1/W2” to represent the answer to the 1st and 2nd bullet points in weaknesses, and “Q1/Q2” to represent the answer to the 1st and 2nd bullet point in questions.
>
>
> > ### W1 (part 1): Intuition about why DiT-based architecture (VEDiT) can be trained with Cross-Entropy loss
>
> Thank you very much for the excellent comment. Intuitively, when we refer to "diffusion", it primarily comprises two parts: (1) a model architecture (e.g., DiT, UNet) used for embedding prediction, and (2) a diffusion/flow-matching scheduler that determines the level of noise (i.e., through different sampling timesteps) during both training and inference.
>
> - For (1), the architecture can technically be any model capable of making predictions by conditioning on different timesteps. As evaluated across several benchmarks mentioned in our main paper, VEDiT is one such model that outperforms previous architectures.
>
> - For (2), using a scheduler with multi-step denoising is one strategy to predict the video embeddings of target clips, often combined with MSE loss during training in previous works like ProceduralVRL [1]. A key innovation and distinction of our method, compared with previous approaches to video embedding prediction for downstream tasks [1, 2]—which typically rely on a combination of multiple loss functions (such as video embedding reconstruction loss and video-language matching loss) for supervision—is that our pipeline can be effectively trained with a single CrossEntropyLoss (instead of the MSE loss mentioned above). Unlike previous works that enforce alignment of the predicted video embeddings with often noisily annotated language descriptions, using a single CrossEntropyLoss allows the optimization target to align more effectively with downstream datasets.
>
> [1] Zhong, Yiwu, et al. "Learning procedure-aware video representation from instructional videos and their narrations.", CVPR 2023
>
> [2] Lin, Xudong, et al. "Learning to recognize procedural activities with distant supervision.", CVPR 2022

---

> ### Author Response · Authors · 2024-11-22
> **Author Response to Reviewer yPQ6 (Part 2/4)**
>
> > ### W1 (part 2): Why we adopt multi-step denoising
>
> Thank you very much for pointing out the valuable comment about the number of transformer blocks passed in each forward pass. Below, we’ll first provide intuitions about why we adopt such a multi-step denoising strategy, then compare the performance of single-step and multi-step denoising to prove it empirically.
>
>
> **1. Intuition about why we adopt multi-step denoising with VEDiT**:
>
> - **1.1 Larger DiTs perform better than smaller DiTs:** One major motivation for us to adopt a deep model (through multi-step denoising) comes from the finding in Diffusion Transformers (DiT) [1]. In Fig. 9 of their paper, they find that with the same amount of training compute (measured by Gflops), larger DiT models are more computationally efficient than smaller DiT models. In addition, small DiT models, even when trained longer, eventually underperform larger DiT models trained for fewer steps.
>
> - **1.2 Multi-step denoising is equivalent to creating a deep model, but in a parameter-efficient way without increasing the number of trainable parameters:** One good property of DiTs, compared with the vanilla transformer models, is that it is able to perform embedding prediction by conditioning on different levels of noises / timesteps. To be specific, DiTs leverage adaptive normalization layers to rescale and shift the input embeddings according to the current time step before passing the embeddings into attention modules (see Fig. 2 “scale, shift, gate” for details). In other words, we can utilize the same model architecture, but run iteratively with different time steps to create a large/deep DiT architecture without multiplying/amplifying the number of trainable parameters  .
>
> - **1.3 Multi-step denoising is more robust compared with single-step denoising:** From 1 and 2 mentioned above, we now know that multi-step denoising can create a “deep” DiT architecture that is proved to be more effective than smaller DiTs. In addition, another good property of multi-step denoising is that it is more robust to single-step denoising. Intuitively, VEDiT denoises the target clip video embedding from random gaussian noise (similar as other denoising models). In each intermediate denoising step, it can freely change the predicted embeddings of target clip based on the information from observed clips. In contrast, single-step denoising does not have such flexibility, and is less robust since it only has a single chance to do prediction from the random gaussian noise.
>
>
> **2. Performance boost via multi-step denoising from empirical results**
>
> On the one hand, in Fig. 6 of our appendix, we have shown that adopting multi-step denoising with a number of denoising steps between 16 and 44 can boost performance compared with single step denoising. On the other hand, we would like to kindly point out that even with single-step denoising, our classification accuracy of **50.77** still outperforms **46.8** from the SOTA baseline (ProceduralVRL) on the step forecasting task. As a reference, ProceduralVRL [2] includes in Table 1 of their paper an “upper bound” of their result by assuming an oracle ranking function that always selects the correct prediction from 5 outputs sampled from their model (i.e., Ours oracle-5). Under such strong assumptions, the performance of this oracle-5 is only 51.8. Therefore, we would like to emphasize that the top-1 classification accuracy improvement from previous SoTA (46.8) to our VEDiT with multi-step denoising (51.8) is non-trivial.
>
>
> [1] Peebles, William, and Saining Xie. "Scalable diffusion models with transformers.", ICCV 2023
>
> [2] Zhong, Yiwu, et al. "Learning procedure-aware video representation from instructional videos and their narrations.", CVPR 2023.

---

> ### Author Response · Authors · 2024-11-22
> **Author Response to Reviewer yPQ6 (Part 3/4)**
>
> > ### W1 (part 3): Other inference/sampling parameters, including denoising steps and classifier-free guidance scales
>
> Thank you very much for this valuable question. Below we answer this question by first clarifying our major difference in inference with respect to traditional image/video generative models, then show some new results with different denoising steps and classifier-free guidance scale.
>
>
> **Our major difference in inference with respect to traditional image/video generative models**
>
> - In traditional image/video generative models, the training and inference process are not directly aligned. For example, during training, only single-step denoising is applied to a randomly sampled noise-level / time step, and no classifier-free guidance (CFG) is applied. During inference, multi-step denoising and (optionally) CFG are applied. In contrast, VEDiT **uses the same aligned strategy during both training and inference**, so the best strategy for us is just to apply the same number of denoising steps and classifier-free guidance scales for inference.
>
> - In addition, we would like to kindly point out that the reason why traditional image/video generation models use CFG is to enable better text-following ability. A larger guidance scale usually leads to better text-following, but also leads to less generated realistic images/videos. So there’s such a tradeoff between visual quality and text-following ability. In contrast, our VEDiT doesn’t have such a tradeoff, with the only goal to predict the video embedding of target clips correctly by conditioning on observed clips. Therefore, it’s not necessary (and probably not optimal) for us to use different CFG scales during training and inference.
>
>
> **Experiment results with different denoising steps**
>
> We would like to kindly emphasize that in appendix Fig. 6, we have shown an ablation of the number of denoising steps. And as we can see, there’s a trend of first increase in classification accuracy as we increase the number of denoising steps, then it starts to decrease as we increase the number of denoising steps more. Therefore, we can see that the number of denoising steps within the range between 20 and 40 can achieve a relative good performance, and we choose 24 in our main paper because it is more computationally friendly compared with more denoising steps.
>
>
> **Experiment results with different classifier-free guidance scale**
>
> Following the Reviewer’s suggestion, we conducted experiments to compare the effect of different CFG scales on the COIN step forecasting and task classification tasks. Specifically, we performed this experiment using the VEDiT architecture with 12 layers and a hidden dimension of 1280, reporting the same top-1 classification accuracy metric used in our main paper.
>
> As shown in the table below, CFG=7 achieves the best step forecasting accuracy, while CFG=3 achieves the best task classification accuracy. Additionally, the accuracy differences across the various CFG scales are relatively small. This is because our model effectively learns parameters that best fit the given CFG scale. Therefore, in the experiments presented in our main paper, we did not fine-tune the CFG parameter and instead used the default CFG value of 7, as in Stable Diffusion 3.
>
>
> ||||
> |-|-|-|
> | | Step Forecasting Accuracy (%) | Task Classification Accuracy (%) |
> | without CFG | 50.73 | 94.70 |
> | CFG = 3 | 50.47 | **94.84** |
> | CFG = 5 | 51.30 | 94.71 |
> | CFG = 7 | **51.85** | 94.67 |
> | CFG = 9 | 51.74 | 94.47 |
> | CFG = 11 | 51.28 | 94.42 |

---

> ### Author Response · Authors · 2024-11-22
> **Author Response to Reviewer yPQ6 (Part 4/4)**
>
> > ### W1 (part 4) & W2: Computational cost of VEDiT
>
> Thank you very much for this question. We have addressed this question in the general comment “Q2: Computational cost of VEDiT”. Please let us know if there are any further concerns about the computational cost of VEDiT. We will do our best to address your concerns and provide detailed illustrations as needed.
>
>
> > ### Q1: Extension to other temporal understanding tasks that predict start/end time
>
> Thank you for the great question! Our model can be adapted to other temporal understanding tasks that involve predicting start and end times. Specifically, we can design the model pipeline similarly to the procedural activity classification task (Fig. 1c in our main paper), with the only difference being the replacement of the attentive classifier, which predicts class labels, with a prediction head that outputs start and end times.
>
> > ### Q2: Performance with different scheduler
>
> Thanks for the great question! Here we first give some explanation about why we choose flow-matching scheduler instead of diffusion (e.g., DDPM), then show quantitative comparison of these two schedulers on step forecasting and task classification tasks.
>
> **Why we use flow-matching sampling**: Flow-matching has been widely used in latest image/video generation works [1, 2] as the default noise scheduler. Compared with traditional diffusion-based noise schedulers, it has a simpler objective that tends to produce a “straighter” flow map, which leads to more robust, more efficient to train, and fewer sampling steps during inference [3, 4].
>
>
> **Model's performance with different scheduler**: We include below the comparison of flow-matching with the widely-used diffusion-based noise scheduler, DDPM [5] on the downstream tasks including COIN step forecasting and task classification. As we can see, compared with DDPM scheduler, Flow-Matching scheduler achieves **1.23%** improvement on step forecasting task and **0.29%** improvement on task classification task, which proves the effectiveness of using Flow-Matching scheduler.
>
>
> ||||
> |-|-|-|
> | | Step Forecasting Accuracy (%) | Task Classification Accuracy (%) |
> | Flow-Matching Scheduler | **51.85** | **94.76** |
> | DDPM Scheduler | 50.62 | 94.47 |
>
>
>
>
>
> [1] Esser, Patrick, et al. "Scaling rectified flow transformers for high-resolution image synthesis.", ICML 2024.
>
> [2] Polyak, Adam, et al. "Movie gen: A cast of media foundation models." arXiv preprint arXiv:2410.13720 (2024).
>
> [3] Ma, Nanye, et al. "Sit: Exploring flow and diffusion-based generative models with scalable interpolant transformers." arXiv preprint arXiv:2401.08740 (2024).
>
> [4] Fischer, Johannes S., et al. "Boosting Latent Diffusion with Flow Matching.", ECCV 2024
>
> [5] Ho, Jonathan, Ajay Jain, and Pieter Abbeel. "Denoising diffusion probabilistic models." NeurIPS 2020
>
>
>
> - - -
> We would like to once more sincerely thank you for all the comments and very useful feedback. We think that we have addressed in depth all Reviewer's questions. If the Reviewer has any additional questions, please let us know, we would be more than happy to answer them.

---

> ### Author Response · Authors · 2024-11-25
> **Official Comment by Authors**
>
> Dear Reviewer yPQ6,
>
> We would like to once more sincerely thank you for your detailed feedback and apologize for taking your time. We believe that we have addressed all your questions, in particular: (1) intuition and detailed discussion about why VEDiT can be effectively trained with Cross-Entropy loss via multi-step denoising, (2) quantitative results about other inference parameters including denoising steps, CFG scales, and different noise scheduler, and (3) computational cost of VEDiT.
>
> We would appreciate your response soon as the deadline for the discussion period ends tomorrow. If our rebuttal is adequate to your questions, we would also appreciate an update in your evaluations. Once more, thank you very much for your reviews and feedback!
>
> Yours sincerely,
>
> The Authors

---

> > ### Comment · Reviewer_yPQ6 · 2024-11-25
> >
> > Thank you for providing such a thorough and detailed rebuttal. I greatly appreciate the effort you put into addressing my concerns and clarifying the key aspects of your work.
> >
> > Your responses have adequately addressed my concerns, and I have updated my score to 6. I would also suggest that the authors consider including the additional experiments and analyses provided in the rebuttal in the main paper.

---

> > > ### Author Response · Authors · 2024-11-26
> > >
> > > Thanks for your positive feedback! We are very glad to hear that your concerns and questions have been adequately-addressed! As suggested, we will include the additional experiments and analyses provided in this rebuttal to the camera-ready version.
> > >
> > > Once more, we sincerely appreciate the time and effort you devoted to reviewing our paper, and we truly appreciate all your valuable comments and suggestions!

---

### Author Response · Authors · 2024-11-22
**General Response**

We sincerely thank all the Reviewers for their valuable feedback and suggestions. We have addressed all of the Reviewers' questions in the individual rebuttals. Below, we provide a summary of the revisions made in our updated PDF, a brief overview of the new experiments, and a general response to common questions raised by the Reviewers.

- - -
**1. Revised contents in the updated PDF:**

Following the suggestions from Reviewer czP8 and Reviewer yPQ6, we have added and refined the following content in our **revised PDF (updated content is highlighted in blue)** to enhance the clarity and comprehensiveness of our paper:

- Distinguished “pretraining on single video clips” from “pretraining on video clip sequences” to avoid ambiguity when stating that our VEDiT does not require large-scale pretraining on procedural video datasets.
- Added a paragraph in the Method section (Sec. 3.1) to explicitly explain our training objective.
- Included a new section (Sec. 5) to discuss the computational cost associated with our multi-step denoising strategy.


- - -
**2. Brief summary of new experiments we conduct following Reviewers’ comments**

In addition, based on the valuable suggestions and comments from the Reviewers, we have conducted several additional experiments that we believe effectively address the Reviewers' concerns and further strengthen our work. Below, we summarize the key experiments and kindly invite the Reviewers to refer to the corresponding individual replies for more details if needed:

- Experiment with different classifier-free guidance scales: see “W1 (part 3): Other inference/sampling parameters, including denoising steps and classifier-free guidance scales” under the response for Reviewer yPQ6.
- Computational cost of VEDiT: see “Q2: Computational cost of VEDiT” under this general response.
- Performance with different noise scheduler: see “Q2: Performance with different scheduler” under the response for Reviewer yPQ6.
- Multimodal Integration: see “W3: Multimodal Integration” under the response for Reviewer Xzeh.
- VEDiT with a simple linear classifier: see “W1: VEDiT with a simple linear classifier” under the response for Reviewer 2SCK.

- - -
**3. General response to common questions from Reviewers**

Again, we would like to sincerely thank all Reviewers for the insightful suggestions. In the following, we address the common questions from the Reviewers as general responses:

---

> ### Author Response · Authors · 2024-11-22
> **Q1: Summary of contributions and novelty of VEDiT**
>
> Reviewer Xzeh and zjEB are concerned about the novelty of our VEDiT. Here, we first describe the motivation of our VEDiT design, then illustrate our differences with the other diffusion-based procedural video literature mentioned by Reviewer Xzeh (i.e., PPDP [1]), and finally our contribution on the reflection of pretraining strategy.
>
> **1. Motivation: Bridging the gap between visual generation and understanding with a diffusion-based architecture**
>
> Recent works on multimodal LLMs have explored the use of unified architectures for both generation and understanding tasks. Building on this trend, we pose the question: _Is it possible to use a diffusion-based generative model architecture for procedural video understanding tasks while maintaining architectural simplicity, achieving strong performance, and enabling extensibility to various procedural video tasks?_ Our work, VEDiT, aims to prove that _a simple generative model architecture, paired with a single task-specific loss function and no additional supervision, can also achieve SoTA performance._
>
> **2. Major differences with the other diffusion-based procedural video learning work (i.e., PPDP)**
>
> Our VEDiT has several major differences that makes it well-distinguished from PPDP:
>
> - **(1) VEDiT works for a wide variety of downstream tasks:** We would like to kindly emphasize that PPDP is specifically designed for the procedural planning task, which predicts intermediate action labels given the start and goal video clips. This limitation stems from its model design, where task class labels, action labels, and video representations are concatenated together as input and output for prediction. In contrast, our VEDiT is designed as a backbone model with a task-specific projection head. _Our design is more flexible, as the backbone model focuses solely on predicting the video embeddings of target clips, while all class label predictions are handled by the task-specific projection head._ This flexibility enables VEDiT to support a wide variety of tasks, including procedural planning, task classification, step forecasting, action grounding, and object state change detection, etc.
>
> - **(2) VEDiT only uses the action labels as supervision**: PPDP is trained with two types of supervision: ground truth task class labels and action labels. In contrast, VEDiT only needs the action labels without any intermediate supervisions such as task class labels.
>
> - **(3) VEDiT is trained in a unified/single stage**: PPDP is trained with two stages. The first stage is to learn the task-related information with the given start and goal video clips. Then the second stage is to generate action sequences with the task-related information and given observations. In contrast, VEDiT is trained in a single stage that directly generates the action sequences.
>
> - **(4) VEDiT is developed based on DiT instead of UNet**: We would like to kindly point out that PPDP uses UNet as the backbone model, while our VEDiT is designed with a bunch of latest techniques in the visual generation community, including DiT-based architecture, RoPE, adaptive normalization layers, and joint attention to fuse the information of observed and target clips.
>
> - **(5) VEDiT achieves SOTA performance with its simple design:** In Table 2 of our main paper, we report that SCEHMA + VEDiT outperforms PPDP and achieves SoTA performance on the procedural planning task. Furthermore, Table 1 shows that VEDiT also surpasses other SoTA models (e.g., ProceduralVRL) on the step forecasting and task classification tasks without relying on large-scale pretraining on massive video datasets (e.g., HowTo100M) or text supervision.
>
>
>
> **3. Our novelty on reflection of pretraining strategy used in previous works**
>
> Our paper also includes discussion of the effectiveness of pretraining strategies used in previous works. We found that the necessity and effectiveness of pretraining the prediction model have not yet been fully validated in previous works for two main reasons.
> - (1) The dominant pretraining objectives (e.g., masked token prediction) were designed for single-clip feature learning, and are not well aligned to the breadth of downstream procedural tasks.
> - (2) Pretraining for sequences rather than single steps demands a scale of data beyond what is currently available. As a result, current approaches fall back on text annotations that are often noisy and poorly temporally aligned with the video content (e.g., ASR narrations).
>
> Therefore, in our work, we investigate how far an approach can go without requiring extensive pretraining._Our use of a frozen encoder combined with a trainable diffusion-based prediction model is not to claim that we are the first to propose such combination, but to show that it offers a powerful alternative to extensive pretraining on massive video datasets._
>
> [1] Wang, Hanlin, et al. "Pdpp: Projected diffusion for procedure planning in instructional videos." CVPR 2023.

---

> ### Author Response · Authors · 2024-11-22
> **Q2: Computational cost of VEDiT**
>
> A common concern (Reviewer yPQ6 and Xzeh) is the lack of computational cost analysis for our VEDiT model in the paper. To address this concern, we first clarify that our goal is not to design a model for real-time inference, as there is typically a tradeoff between efficiency and performance. Additionally, we include new experiments comparing training and inference clock time and GPU memory usage, which demonstrate that VEDiT with $N$-step denoising does not result in $N$ times longer training and inference times.
>
> **1. Many procedural video tasks do not require real-time inference.**
>
> Enabling real-time inference is a parallel topic to our paper, which usually involves model distillation, quantization, and optimization on the hardware levels. And actually, many strong baselines on procedural video understanding tasks even involve calling heavy LLMs or MLLMs [1, 2, 3], which could be much more computationally costly compared with our model.
>
>
> **2. There’s usually a tradeoff between efficiency and performance.**
>
> In Fig. 6 of our appendix, we have shown that adopting multi-step denoising with a number of denoising steps between 16 and 44 can boost performance compared with single step denoising. On the other hand, we would like to kindly point out that even with single-step denoising, our classification accuracy of **50.77** still outperforms **46.8** from the SOTA baseline (ProceduralVRL) on the step forecasting task. As a reference, ProceduralVRL [2] includes in Table 1 of their paper an “upper bound” of their result by assuming an oracle ranking function that always selects the correct prediction from 5 outputs sampled from their model (i.e., Ours oracle-5). Under such strong assumptions, the performance of this oracle-5 is only 51.8. Therefore, we would like to emphasize that the top-1 classification accuracy improvement from previous SoTA (46.8) to our VEDiT with multi-step denoising (51.8) is non-trivial.
>
> **3. New experiments comparing the training and inference clock time and GPU memory**
>
> That being said, we evaluated the training and inference time, as well as GPU memory usage, of our model with varying numbers of denoising steps. Specifically, we conducted this experiment using the VEDiT architecture with 12 layers and a hidden dimension of 1280 (696M trainable parameters). We measured the clock time and GPU memory required to run inference on the model using 1 COIN video consisting of 8 clips, with gradient checkpointing enabled.
>
> As shown in the table below, increasing the number of denoising steps from 1 to 24 results in only a 1.32x increase in inference time and a 2.02x increase in training time. This efficiency is partially because of the adoption of scalable dot product attention in each VEDiT block. Moreover, because we employ gradient checkpointing to optimize GPU memory usage (at the cost of additional training and inference time), GPU memory usage remains nearly constant without significant variation.
>
> Thus, we would like to kindly highlight that _the training and inference times only increase by a factor of 1-2x as the number of denoising steps increases from 1 to 24, rather than scaling linearly with the number of steps (e.g., 24x)_. This demonstrates that the efficiency of our model is not a bottleneck for its practical use.
>
>
>
> |||||||||
> |-|-|-|-|-|-|-|-|
> | # Denoising Steps | 1 | 4 | 8 | 12 | 16 | 20 | 24 |
> | Inference Clock Time (second) | 0.40 | 0.47 | 0.56 | 0.66 | 0.75 | 0.84 | 0.93 |
> | Inference GPU Memory (GB) | 15.7 | 15.7 | 15.8 | 15.8 | 15.8 | 15.8 | 15.9 |
> | Training Clock Time (second) | 0.41 | 0.49 | 0.63 | 0.82 | 0.94 | 1.09 | 1.24 |
> | Training GPU Memory (GB) | 21.8 | 21.9 | 22.0 | 22.0 | 22.1 | 22.2 | 22.3 |
>
>
>
>
> [1] Zhao, Qi, et al. "AntGPT: Can Large Language Models Help Long-term Action Anticipation from Videos?." ICLR 2024
>
> [2] Pei, Baoqi, et al. "Egovideo: Exploring egocentric foundation model and downstream adaptation." arXiv 2024
>
> [3] Huang, Daoji, et al. "Palm: Predicting actions through language models@ ego4d long-term action anticipation challenge 2023." arXiv 2023

---

### Meta-Review · Area_Chair_m1pK · 2024-12-05

**Metareview:**

This paper receives final scores of 6,6,6,6,6, from five reviewers. Reviewers have raised several issues on the architectural novelty, scaling, and lack of detailed experimental analysis. The authors have addressed most of the concerns in the rebuttal. Thus, there is a consensus of acceptance. AC has checked the submission, the reviewers, and the rebuttal and agreed with the consensus of all five reviewers. Thus acceptance is recommended.

**Additional Comments On Reviewer Discussion:**

The authors have run many experiments and updated the submission to address most of the concerns of the five reviewers in the discussion.

---

### Decision · Program_Chairs · 2025-01-22

Accept (Poster)